# Influence of Photochemical Loss of VOCs on Understanding Ozone Formation Mechanism

Wei Ma[1], Zemin Feng[1,7], Junlei Zhan[1], Yongchun Liu[1*], Pengfei Liu[2,4,5], Chengtang Liu[2,4,5], Qingxin

Ma[2,4,5], Kang Yang[3], Yafei Wang[3], Hong He[2,4,5], Markku Kulmala[1,6], Yujing Mu[2,4,5], Junfeng Liu[2,4,5*]

1. Aerosol and Haze Laboratory, Advanced Innovation Center for Soft Matter Science and Engineering,

Beijing University of Chemical Technology, Beijing, 100029, China

2. Research Center for Eco-Environmental Sciences, Chinese Academy of Sciences, Beijing, 100085, China

3. Beijing Institute of Petrochemical Technology, Beijing 102617, China

4. Center for Excellence in Regional Atmospheric Environment, Institute of Urban Environment, Chinese

Academy of Sciences, Xiamen 361021, China

5. University of Chinese Academy of Sciences, Beijing 100049, China

6. Institute for Atmospheric and Earth System Research, Faculty of Science, University of Helsinki, Helsinki,

00014, Finland

7. College of Chemical Engineering, North China University of Science and Technology, Tangshan 063021,

China.

*Corresponding to: Yongchun Liu (liuyc@buct.edu.cn) and Junfeng Liu (junfengliu@rcees.ac.cn)

## Abstract

Volatile organic compounds (VOCs) tend to be consumed by atmospheric oxidants, resulting in substantial photochemical loss during transport. An observation-based model was used to evaluate the influence of photochemical loss of VOCs on the sensitivity regime and mechanisms of ozone formation. Our results showed that a VOC-limited regime based on observed VOC concentrations shifted to a transition regime with a photochemical initial concentration of VOCs (PIC-VOCs) in the morning. The net ozone formation rate was underestimated by 3 ppb h$^{-1}$ (~36 ppb day$^{-1}$) based on the PIC-VOCs. The relative contribution of the RO$_2$ path to ozone production based on the PIC-VOCs accordingly increased by 13.4%; in particular, the contribution of alkene-derived RO$_2$ increased by approximately 10.2%. In addition, the OH-HO$_2$ radical cycle was obviously accelerated by highly reactive alkenes after accounting for photochemical loss of VOCs. The contribution of local photochemistry might be underestimated for both local and regional ozone pollution if consumed VOCs are not accounted for, and policymaking on ozone pollution prevention should focus on VOCs with a high reactivity.

# 1. Introduction

Ground surface ozone ($O_3$) is an important atmospheric pollutant that is harmful to human health and is connected with respiratory, cardiovascular diseases, and premature mortality (Cohen et al., 2017). It is also harmful to vegetation growth. For example, it led to annual reductions in the yields of rice and wheat by 8% and 6%, respectively, and reduced forest biomass growth by 11-13% in China (Feng et al., 2019). Surface $O_3$ concentrations have increased by 11.9% over eastern China despite the air pollution control measures implemented in China from 2012 to 2017 (Dang and Liao, 2019). An economic loss of 0.09% of the Chinese gross domestic product (78 billion CNY) is predicted for 2030 if policies against $O_3$ pollution are not properly implemented (Xie et al., 2019). Therefore, urgent action to minimize $O_3$ pollution in China is needed.

Tropospheric $O_3$ is mainly produced from photochemical reactions between volatile organic compounds (VOCs) and nitrogen oxides (NO$x$: NO+NO$_2$) (Seinfeld and Pandis, 2006; Liu et al., 2021). $O_3$ is generated from a collision of $O_2$ and O(3P) that is produced from photolysis of $NO_2$ in the atmosphere. Peroxyl radicals ($HO_2$ and $RO_2$), which are produced from the oxidation of VOCs by OH radical, can efficiently convert NO (from the photolysis of $NO_2$) to $NO_2$, leading to a net $O_3$ production by compensating for the titration of $O_3$ by NO (Monks, 2005; Zhang et al., 2021a). Over the past two decades, a number of field observations focused on $O_3$ pollution levels and its precursors have been carried out in the Beijing-Tianjin-Hebei (BTH), Yangtze River Delta (YRD), and Pearl River Delta (PRD) regions (Wang et al., 2017; Li et al., 2019; Xue et al., 2014; Zhang et al., 2019). Due to the nonlinear relationship

between $O_3$ and its precursors and the variations in meteorological conditions, numerous
studies have been performed to understand the sensitivity regime of $O_3$ formation (Ling and
Guo, 2014; Zhang et al., 2020), the photochemical process of $O_3$ formation based on box
models or observation-based models (OBM) (He et al., 2019; Tan et al., 2019), and the sources
of $O_3$ using regional chemical transport models (Li et al., 2016b; Li et al., 2016c). Recently,
the instantaneous production rate of the $O_3$ formation process has attracted more attention; for
example, studies examining radical recycling (OH-$RO_2$-RO-$HO_2$-OH) related to the production
of $O_3$ have been performed (Lu et al., 2017; Tan et al., 2017; Whalley et al., 2018). HCHO
photolysis and alkene ozonolysis contributed approximately 85% to the primary production of
$HO_2$ and HO radicals in Beijing, Shanghai and Guangzhou (Tan et al., 2019; Yang et al., 2017).
The importance of HONO and HCHO photolysis for primary radical production has also been
proposed in suburban and rural areas (Tan et al., 2017; Lu et al., 2012; Lu et al., 2013).

67        All of the OBM studies investigating the relationship between $O_3$ and VOCs were based

on measured datasets. However, VOCs are highly reactive to atmospheric oxidants, such as
OH, $NO_3$, and $O_3$, among which OH is dominant. The lifetimes of some highly reactive VOCs,
such as isoprene, are as short as only a few tens of minutes under typical daytime atmospheric
conditions. The mixing ratios of VOCs observed at a sampling site are actually the residues of
VOCs from emissions due to the photochemical loss during transport from the source site to
the receptor site. If photochemically consumed VOCs are not considered, the $O_3$ formation
sensitivity and net $O_3$ production may be misunderstood, and subsequent policymaking on $O_3$
pollution prevention at regional or urban scales may be misguided. Thus, the photochemical
age-based approach has been applied to evaluate the effect of photochemical processes on VOC
measurements (Shao et al., 2011). This method was used to qualitatively or semi-quantitatively
estimate the $O_3$ formation process of the source-receptor (Gao et al., 2018) by calculating the
$O_3$ formation potential (OFP) (Han et al., 2017), identifying the critical species for $O_3$ formation
(Gao et al., 2021), or evaluating the VOC emissions ratio (Yuan et al., 2013). In evaluating the
importance of initial VOCs to ozone production, Xie et al. (2008) found that the OFP at a
Peking University site increased by 70% after accounting for the photochemical loss of VOCs.
Li et al. (2015) also showed that the OFPs of total NMHCs (excluding isoprene) increased by
16.1% (from 59.6 to 69.2 ppb $O_3$), 12.1% (from 33.5 to 37.5 ppb $O_3$), and 3.4% (from 68.9 to
71.2 ppb $O_3$) after correcting for photochemical loss in Gucheng, Quzhou, and Beijing,
respectively. Gao et al. (2018) reported that the OFP could be underestimated by 23.4% (62.4
ppb $O_3$) in Beijing if the photochemical loss of VOCs is not considered. Zhan et al. (2021)
found that based on measured VOCs, the OFP increased from 57.8 ppb to 103.9 ppb using the
initial VOCs. All the previous work was based on the maximum incremental reactivities (MIR)
method. The application of such calculations using the MIR method is restricted to areas or
episodes in which $O_3$ formation is VOC-sensitive (Carter, 1994). In the troposphere, the
sensitivity of ozone formation to NO$x$ and VOCs varies greatly, as evidenced by the wide range
of OFP underestimations from ~3% to 70% in previous work. In addition, the MIR values of
VOC species for a specific region are calculated with the base scenario, in which NO
concentration and other parameters are the values that correspond to the maximal incremental
reactivity (IR). The fixed MIR values of different VOCs can neither reflect the non-linear

relationship between ozone and VOCs, involving in the complicated radical recycling (OH-RO$_2$-RO-HO$_2$-OH) related to the production of ozone, nor be used for analyzing the radical budget of the initial VOCs concentration. Thus, a quantitative analysis is necessary to explicitly understand the influence of photochemical loss of VOCs on ozone formation and its mechanism based on OBM studies, in which the dynamic atmospheric and meteorological conditions is accounted for.

In this study, an OBM was used to evaluate the local O$_3$ formation process in summer in Beijing based on concentrations of observed and photochemical initial concentrations of VOCs (PIC-VOCs). The O$_3$-NO$x$-VOC sensitivity, instantaneous O$_3$ formation rate and in situ O$_3$ formation process were discussed. The aim of this study was to understand the possible influence of photochemical loss of VOCs on the formation sensitivity regime of O$_3$ and how the photochemical loss of VOCs affects O$_3$ formation. This study can provide new insight for better understanding atmospheric O$_3$ pollution.

## 2. Methodology

### 2.1 Experimental section

Field observations were carried out on the Qingyuan campus of the Beijing Institute of Petrochemical Technology (BIPT, 39.73°N and 116.33°E) (Figure S1). Details on the observation site have been described in our previous work (Zhan et al., 2021). In short, the site is a typical suburban site in the Daxing District between 5[th] Ring Road and 6[th] Ring Road. The field campaign was carried out during August 1-28, 2019, when photochemistry was the most active and rainfall was rare in Beijing.

The concentrations of nonmethane hydrocarbons (NMHCs) were detected by both a gas
chromatography-flame ionization detector (GC/FID) and a single photon ionization (SPI) TOF-
MS (SPI-MS 3000, Guangzhou Hexin Instrument Co., Ltd., China). A detailed description of
the instrumentation can be found in previous publications (Zhan et al., 2021; Chen et al., 2020).
The SPI-MS was also used to detect halohydrocarbons. More details on this instrument and its
parameter settings have been described in previous studies (Zhang et al., 2019; Liu et al.,
2020a). In short, a 0.002 int thick polydimethylsiloxane (PDMS) membrane (Technical
Products Inc., USA) was used to collect VOCs and diffuse them from the sample site to the
detector under high vacuum conditions. Vacuum ultraviolet (VUV) light generated by a
commercial D2 lamp (Hamamatsu, Japan) was utilized for ionization at 10.8 eV. For ion
detection, two microchannel plates (MCPs, Hamamatsu, Japan) assembled with a chevron-type
configuration were employed. This TOF-MS has an LOD varying from 50 ppt to 1 ppb with a
1-minute time resolution for most trace gases without any preconcentration procedure. To
verify the data compatibility of the SPIMS and GC/FID, we compared the concentrations of
toluene measured using these two different instruments (Figure S2). The correlation coefficient
was 0.9 (with a slope of 0.7), indicating that the concentrations of NMHCs were comparable
using these two measurement techniques.
Oxygenated VOCs (OVOCs) were collected using 2,4-dinitrophenylhydrazine (DNPH)-
coated silica gel cartridges (Sep-Pak, Waters) by an automatic sampling device with a sampling
flow rate of 1.2 L min$^{-1}$ and a duration of 2 h for each sample. Then, the OVOCs were analysed
using high-performance liquid chromatography (HPLC, Inertsil ODS-P 5 μm 4.6 × 250 mm
column, GL Sciences) with an acetonitrile-water binary mobile phase (Ma et al., 2019). To
avoid possible contamination or desorption after sampling, cartridges were capped, placed into
tightly closed plastic bags and kept in a refrigerator before analysis. The sampled cartridges
were eluted with 5 mL acetonitrile and analysed by HPLC as soon as possible after they were
shipped back to the laboratory. This system was calibrated with 8-gradient standard solutions
(TO11/IP-6A Aldehyde/Ketone-DNPH Mix, SUPELCO). The correlation coefficients were all
greater than 0.999. The LOD for most OVOCs was approximately 10 ppt.
Trace gases, including $NO_x$, $SO_2$, CO, and $O_3$, were measured using corresponding
analysers (Thermo Scientific, 42i, 43i, 48i, and 49i, respectively). The HONO concentration
was measured using a homemade long path absorption photometer (LOPAP) (Liu et al., 2020c).
The meteorological parameters, including temperature (T), pressure (P), relative humidity
(RH), wind speed, and wind direction, were measured by a weather station (AWS310, Vaisala).
The photolysis rate ($J_{NO2}$) was measured via continuous measurement of the actinic flux in the
wavelength range of 285-375 nm using a $J_{NO2}$ filter radiometer ($J_{NO2}$ radiometer, Metcon).
**2.2 Calculation of photochemical loss of VOCs**
The photochemical loss of VOCs was calculated using the ratio method (Wiedinmyer et
al., 2001; Yuan et al., 2013). The initial mixing ratio of a specific VOC was calculated using
the following equations (Mckeen et al., 1996):
$$[VOC_i]_t = [VOC_i]_{t0} \times \exp(-k_i \times [OH] \times \Delta t) \quad (1)$$

$$\Delta t = \frac{1}{[OH] \times (k_X - k_E)} \times \left\{ ln\left(\frac{X_0}{E_0}\right) - ln\left(\frac{X_t}{E_t}\right) \right\} \quad (2)$$

where $[VOC_i]_t$ and $[VOC_i]_{t0}$ are the observed and initial concentrations of $VOC_i$, respectively;
$k_i$ is the second-order reaction rate between compound $i$ and OH radical; and [$OH$] and $\Delta t$ are
the concentration of OH radical and the photochemical ageing time, respectively. $k_X$ and $k_E$ are
rate constants for the reaction between OH radicals and ethylbenzene ($7.00 \times 10^{-12}$ cm$^3$
molecule$^{-1}$ s$^{-1}$) and xylene ($1.87 \times 10^{-11}$ cm$^3$ molecule$^{-1}$ s$^{-1}$) (Atkinson and Arey, 2003),
respectively. ($X_0/E_0$) is the initial mixing ratio between xylene and ethylbenzene, and ($X_t/E_t$) is
the mixing ratio between xylene and ethylbenzene at the observation time. In this study, we
chose the mean concentrations of xylene and ethylbenzene at 05:00-06:00 as their initial
concentrations before sunrise according to the ambient $J_{NO2}$ (Figure S3) to calculate the
photochemical loss of OH exposure. In previous work (Shao et al., 2011; Zhan et al., 2021),
the selection of ethylbenzene and xylene as tracers was justified for calculating ambient OH
exposure under the following conditions: 1) the concentrations of xylene and ethylbenzene
were well correlated (Figure S4), which indicated that they were simultaneously emitted; 2)
they had different degradation rates in the atmosphere; and 3) the calculated PICs were in good
agreement with those calculated using other tracers, such as i-butene/propene (Figure S5)
(Zhan et al., 2021). To test the relative constant emission ratio from different sources, we chose
benzene vs. acetylene and n-hexane vs. toluene as references, and the result is shown in Figure
S6. These ambient ratios could directly reflect their relative emission rates from sources
(Goldan et al., 2000; Jobson et al., 2004). The linear correlation coefficients ($R^2$) were generally
higher than 0.7, which were equal to that reported by Shao et al. (2011). To further test the
assumption that the emissions of xylene and ethylbenzene were constant throughout the day,
their potential sources were calculated using a source-receptor model (the potential source
contribution function, PSCF). As shown in Figure S7, xylene and ethylbenzene showed similar
distributions. In addition, the ratio of ethylbenzene/xylene at 5:00 and 6:00 was similar to that
during the daytime. These results indicated that the emissions of xylene and ethylbenzene were
constant throughout the day. The ratio of xylene to ethylbenzene and the OH exposure
concentration are shown in Figure S8. The results showed that the ratio of xylene to
ethylbenzene increased gradually (07:00~12:00), which is consistent with the trend of xylene
and ethylbenzene. The OH exposure was from 0.82 to $8.1\times10^6$ molecule $cm^{-3}$ h, with a mean
daytime value of $4.3\pm1.9\times10^6$ molecules $cm^{-3}$ h. Accordingly, the mean photochemical ages
were $1.7\pm0.9$ h using the mean daytime (8:00-17:00 LT) OH concentrations ($4.3\pm3.1\times10^6$
molecules $cm^{-3}$) calculated based on JO1D using the method reported in our previous work
(Liu et al., 2020b; Liu et al., 2020c). This meant that VOCs would undergo obvious degradation
even during a short range of transport in the atmosphere.

193        It should be noted that the $k_{OH}$ of isoprene is $9.98\times10^{-11}$ $cm^3$ $molecule^{-1}$ $s^{-1}$ at 298.15 K

(Atkinson and Arey, 2003), almost two orders of magnitude greater than other VOCs. The ratio
method assumes constant emissions for VOCs. However, the emission of isoprene greatly
depends on temperature and solar irradiation intensity (Zhang et al., 2021b). In addition to
accounting for photochemical loss, additional correction of daytime isoprene concentrations
was performed using the average diurnal flux of isoprene emissions (Figure S9 (Zhang et al.,
2021b). The emission of isoprene showed a clear unimodal curve, and the volume
concentration of isoprene was calculated based on the daily emission curve using Eq. (S1).

**2.3 Observation-based model simulation**

A box model based on the Master Chemical Mechanism (MCM3.3.1) and the Regional Atmospheric Chemical Mechanism (RACM2) was used in this study. The MCM3.3.1 was used to understand the instantaneous ozone formation process, and the RACM2 was used to depict the ozone isopleth due to its high computational efficiency (Sect. 2.4). Table S1 shows the model inputs. The model calculations were constrained with the measured meteorological parameters (RH, T, P, and $J_{NO2}$) and the concentrations of trace gases, including inorganic species (NO, $NO_2$, CO, $SO_2$, and HONO) and 61 organic species (NMHCs (46), OVOCs (8), and halohydrocarbons (7)). The model was validated using the observed and simulated $O_3$ concentrations, which showed good consistency, as shown in Figure S10. The slope and correlation coefficients were 0.9 and 0.8, respectively (Figure S11, respectively, indicating the validity of the model simulation. It is worth mentioning that the results of model simulation can sometimes be overestimated or underestimated to some extent, which has also been reported by previous studies (Zong et al., 2018; Zhang et al., 2020), but this did not affect our simulations of the ozone formation process and mechanisms because we constrained the ozone concentration during our simulations.

The ozone formation rate $P(O_3)$ can be quantified by the oxidation rate of NO to $NO_2$ by peroxyl radicals (Tan et al., 2019), as expressed in Eq. (3). In this study, the modelled peroxyl radical concentrations were used to calculate the ozone production rate.

$$P(O_3) = k_{HO_2+NO}[HO_2][NO] + k_{RO_2+NO}[RO_2][NO] \quad (3)$$

where $P(O_3)$ is the ozone formation rate; $[HO_2]$ and $[RO_2]$ are the number concentrations of

$HO_2$ and $RO_2$ radicals; $k_{HO2+NO}$ is the second reaction rate between $HO_2$ and $NO$; and $k_{RO2+NO}$
is the second reaction rate for the reaction of $RO_2$ and $NO$, which only produces $RO$ and $NO_2$.
Once ozone forms, it will be consumed by $OH$, $HO_2$, and alkenes. Additionally, some $NO_2$ can
react with $OH$, resulting in the formation of nitrate before photolysis. The chemical loss of both
$O_3$ and $NO_2$ is considered in the calculation of the net ozone production rate (Tan et al., 2019),
$$L(O_3) = \left(k_{O_3+OH}[OH] + k_{O_3+HO_2}[HO_2] + k_{O_3+alkenes}[alkenes]\right)[O_3] +$$
$$k_{NO_2+OH}[NO_2][OH] \quad (4)$$
where $L(O_3)$ is the ozone chemical loss rate; $[OH]$ is the number concentration of $OH$ radical;
$k_{O3+OH}$, $k_{O3+HO2}$, and $k_{O3+alkenes}$ are the second-order reaction rate constants between $O_3$ and $OH$,
$HO_2$ and alkenes, respectively; and $k_{NO2+OH}$ is the second-order reaction rate constant between
$NO_2$ and $OH$. Finally, $F(O_3)$ is the net ozone formation rate calculated by the difference between
$P(O_3)$ and $L(O_3)$, as expressed in Eq. (5),
$$F(O_3) = P(O_3) - L(O_3) \quad (5)$$
**2.4 Empirical Kinetic Modelling Approach**
The empirical kinetic modelling approach (EKMA) used in this work is a set of imaginary
tests to reveal the dependence of photochemical oxidation products on the change in precursors.
We set up 30 × 30 matrices by reducing or increasing the measured VOCs and NO$x$
concentrations in the model input. The resulting radical concentrations and ozone production
rates were calculated correspondingly.
At this stage, the observed VOCs were grouped into different lumped species according
to their RACM2 classification; more details can be found in a previous publication (Tan et al.,
2017). The chemical model simulated photochemical reactions with input species for a time
interval of 60 minutes, which was enough for $NOx$, $OH$, $HO_2$, and $RO_2$ to reach a steady state
because the typical relaxation time of the chemical system is 5-10 minutes in summer (Tan et
al., 2018). However, all the species and parameters were input at a 5 min interval by data
interpolation to reduce simulation inconsistencies and large distortions of meteorological
parameters at longer time intervals (Tan et al., 2018). The ozone production rate was calculated
as described in Sect. 2.3. It is worth mentioning that the average survey data were selected as
the baseline scenario in simulating the EKMA curve in this study.

## 251 3. Results and discussion

### 252 3.1 Overview of diurnal variation in $O_3$, $NOx$, and TVOC

Figure 1 shows the average diurnal variation of concentrations in $O_3$, $NOx$, and TVOC
(including alkanes, alkenes, OVOCs, and halohydrocarbons) driven by emissions,
photochemical reactions and the evolution of the mixing layer height (MLH). The ozone
concentration during the observation period was 44.8±27.2 ppb with a maximum of 119.1 ppb,
as reported in our previous study (Zhan et al., 2021), which was generally comparable with the
$O_3$ concentrations during 2014-2018 (Ma et al., 2020). The $O_3$ followed a unimodal curve with
a minimum value (18.8±15.4 ppb) at 07:00 and then it increased to a maximum value (69.6
ppb) at 15:00 as photochemical ozone formed. In contrast, $NOx$ reached its maximum
concentration (39.7±14.2 ppb) at 07:00 and then decreased. After 07:00, the mixing ratio of
NO continuously dropped, while the concentration of $NO_2$ decreased at first and then started
to increase at 14:00. The diurnal variations in the observed TVOCs were generally consistent
with those of NO₂. The observed TVOCs concentrations ranged from 2.2 to 23.2 ppb, with a
mean value of 18.6±2.6 ppb. Compared to the concentrations (45.4±15.2 ppb) in the same
period in August 2015 (Li et al., 2016a), the concentration of VOCs in Beijing was effectively
reduced. However, the photochemical initial concentrations (PICs) of TVOCs, which varied
from 2.2 to 27.8 ppb with a mean value of 24.5±2.1 ppb, showed a different diurnal curve
compared with the observed concentrations. It slightly increased from 07:00 to 14:00, which
was similar to the diurnal variation of VOCs in previous work (Zhan et al., 2021). The average
PIC-VOCs was 6.9±0.5 ppb higher than the observed concentration of TVOCs, indicating an
underestimated contribution of the local photochemistry of VOCs to $O_3$ and organic aerosol
formation.

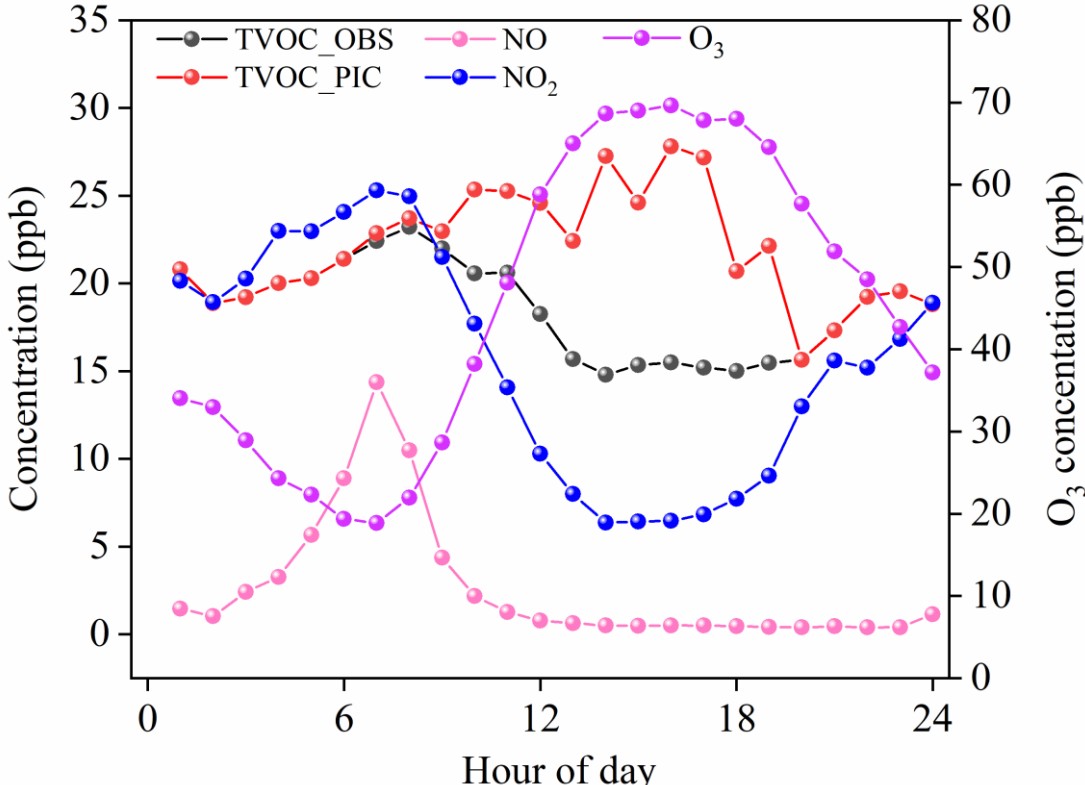


**Figure. 1.** Overview of average diurnal variations in $O_3$, NO$x$, and TVOC. The data represent
measured results, except for those of the TVOC_PIC, which are calculated based on OH radical
exposure. The data range is August 1-28, 2019.

**3.2 Influence of photochemical loss of VOCs on the O$_3$ formation sensitivity regime**

The sensitivity of O$_3$ formation is analysed using the isopleth diagram generated from the
EKMA model, which is widely used to qualitatively study O$_3$-NO$x$-VOCs sensitivity. As
described in Sect. 2.4, the concentrations of NO$_2$ and VOCs were artificially scaled to ±75%
of the observed values to calculate the response of O$_3$ concentration to an imaginary change in
the concentrations of NO$_2$ and VOC, with other constrained conditions remaining unchanged.
Figure 2 shows the typical EKMA curves during our observations. The black stars and
pentagons denote the observed concentrations of NO$x$ and VOCs in the morning (09:00-10:00)
and at noon (14:00-15:00), respectively, while the blue symbols are the corresponding values
of PICs. Based on the measured data, O$_3$ formation was in a VOC-limited regime in the
morning and a NO$x$-limited regime in the afternoon. The black arrow indicates a linearly
decreasing trend of NO$x$ and VOCs from 09:00 to 15:00 in the chemical coordinate system,
and ozone production shifted from VOC-limited to NOx-limited conditions from morning to
afternoon, which was consistent with the mean diurnal profiles (Figure 1). This was similar to
the data reported in Wangdu (Tan et al., 2018). As expected, ozone production shifted from a
VOC-limited regime (the observed VOCs) to a transition regime based on the PIC-VOCs in
the morning. Ozone production clearly moved further to a NO$x$-limited regime in the afternoon
after the photochemically consumed NO$x$ and VOCs had been accounted for (Figure 2).
Because the average photochemical ageing time was only 1.7±0.9 h, these results indicated
that the O$_3$ formation mechanism might typically be misdiagnosed, which misleads mitigation
measures for $O_3$ prevention if the consumed VOCs under real atmospheric conditions are not
considered.

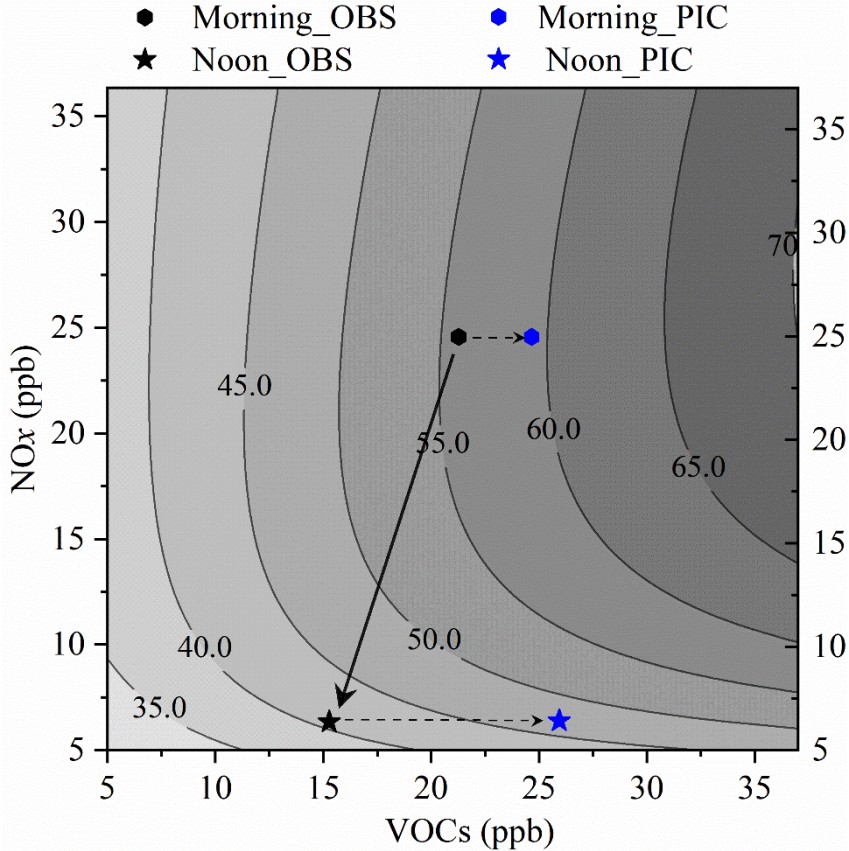


**Figure 2.** Isopleth diagram of the ozone concentration as a function of the concentration of
NO$x$ and VOCs derived from an empirical kinetic modelling approach. The pentagons and stars
indicate the status in the morning (09:00-10:00) and at noon (14:00-15:00), respectively. The
black and blue colours represent the observed and corrected statuses, respectively.
**3.3 Contribution of VOC species to O$_3$ production**

306         The time series of simulated OH, HO$_2$, and RO$_2$ concentrations were used to calculate the

$P(O_3)$ and $L(O_3)$. The diurnally averaged $P(O_3)$ and $L(O_3)$ are shown in Figure 3. Ozone
formation can be divided into processes related to RO$_2$+NO and HO$_2$+NO (Sect. 2.3).
According to their VOC precursors, peroxyl radical groups were divided into alkane-derived
(ALKAP), alkene-derived (ALKEP), aromatic-derived (AROMP), isoprene-derived (ISOP),
oxygenated-VOC-derived (OVOCP), and halohydrocarbon-derived (HALOP) $RO_2$ and HO2P.
The ozone destruction processes included the reaction between $O_3$ and HO$x$ (O3D1), the
reaction between O1D and $H_2O$ (O3D2), the reaction between $O_3$ and alkenes (O3D3), and the
reaction between $NO_2$ and OH (O3D4).

315        Based on the observed VOCs (or PIC-VOCs), a fast $O_3$ production rate was observed at

14:00 (or 13:00), with a diurnal maximum value of 16.1 (or 25.6) ppb $h^{-1}$ (Figure 3a and 3b),
while the peak destruction rate was 6.4 (or 8.6) ppb $h^{-1}$ at 15:00 (or 13:00) (Figure 3c and 3d).
The average daytime $P(O_3)$ from 07:00 to 19:00 based on the initial concentrations of VOCs
was $4.0\pm3.1$ ppb $h^{-1}$ higher than that based on the measured VOCs concentrations (Figure 3b).
At the same time, the $F(O_3)$ from 07:00 to 19:00 based on the initial concentrations of VOCs
was also $3.0\pm2.1$ ppb $h^{-1}$ higher than the measured counterpart (Figure S12). Thus, the net $O_3$
production could be accumulatively underestimated by ~36 ppb $day^{-1}$ from 07:00 to 19:00 if
the consumption of VOCs was not considered. This meant that the contribution of the local
formation of $O_3$ could be underestimated using the directly measured VOCs concentrations. It
should be pointed out that it is better to compare $O_3$ production with the true metric for $O_3$
production. However, it is impossible to directly measure the true metric for $O_3$ production in
the atmosphere at the present time to know how well the method presented here corrects for
that underestimation. In addition, the ozone concentrations must be constrained when
simulating the ozone formation process (Lu et al., 2013; Tan et al., 2017). Thus, it is impossible
to directly compare the ozone production based on PIC-VOCs with that using measured VOCs
concentrations. Therefore, we alternatively compared the integrated net ozone production rates
rather than ozone production or concentrations between the two scenarios. An upwind $O_3$ and
VOCs measurement combined with a trajectory analysis might provide an approach for
checking the accuracy of our results. Alternatively, conducting a transient $O_3$ production rate
analysis after subtracting the transport of $O_3$ with a regional model and/or satellite observation
might be another option. Unfortunately, neither the upwind measurement nor the regional
model simulation was available at the time of our study. To further check the accuracy of our
results, we chose August 4[th] as a test case to explore the influence of the transport of ozone on
a downwind site based on the trajectory analysis. As shown in Figure S13, the mean ozone
concentration of the downwind site (national monitoring station, NMS) was 27.6±21.9 ppb
day$^{-1}$ higher than that of the observation site (OS), which was slightly less than the difference
(~36 ppb day$^{-1}$) between PIC-VOCs and observed VOCs and indirectly rationalized our results.

343        The $HO_2$ path contributed 64.8% to the total ozone formation on average, which was

slightly higher than the reported value (57.0%) in Wangdu (Tan et al., 2018), whereas the $RO_2$
path, in which aromatics (9.4%), alkenes (8.4%), isoprene (7.8%), alkanes (4.7%), OVOCs
(4.3%) and halohydrocarbons (0.6%) were the main contributors, contributed to the remaining
part. For the PIC-VOCs, the dominant path of $O_3$ production (51.7%) was still the $HO_2$ path,
followed by the $RO_2$ path related to alkenes (14.7%), aromatics (12.8%), and isoprene (11.7%).
The relative contribution of the $RO_2$ path to $P(O_3)$ increased by 13.4% compared with the
measured VOCs, particularly alkene-derived $RO_2$, which increased by 10.2%. As shown in
Figure 3c and 3d, the destruction of total oxidants was dominated by the reaction between $O_3$
and alkenes (O3D3) in the morning. It gradually shifted to the reaction between $NO_2$ and OH
(O3D4) from 11:00 to 16:00 and the photolysis of $O_3$ followed by a reaction with water (O3D2)
from 12:00 to 15:00 because $O_3$ concentration increased while $NO_2$ decreased (Figure 3c).
Figure S14 shows the percentages of the different paths of $P(O_3)$ and $L(O_3)$. The relative
contributions of the reactions between $O_3$ and alkenes (O1D3) and between $NO_2$ and OH
(O1D4) to the $O_3$ sinks decreased when calculated based on PIC-VOCs compared with those
of the measured VOCs, while they obviously increased for the other two paths, i.e., O3D1 and
O3O2. The $O_3$ destruction of the HO$x$ and $O_3$ reaction (O3D1) gradually increased with the
continuous photochemical reaction. In addition, the maximum $O_3$ formation rates of the $RO_2$
derived from OVOCs and halohydrocarbons were 0.75 and 0.18 ppb $h^{-1}$, respectively. These
values could be underestimated due to the incomplete gas reaction mechanism of OVOCs and
halohydrocarbons in MCM3.3.1. In general, the measured VOCs as model inputs could fail to
truly reflect the oxidation capacity and underestimate the local formation of $O_3$ and organic
aerosols (Zhan et al., 2021).

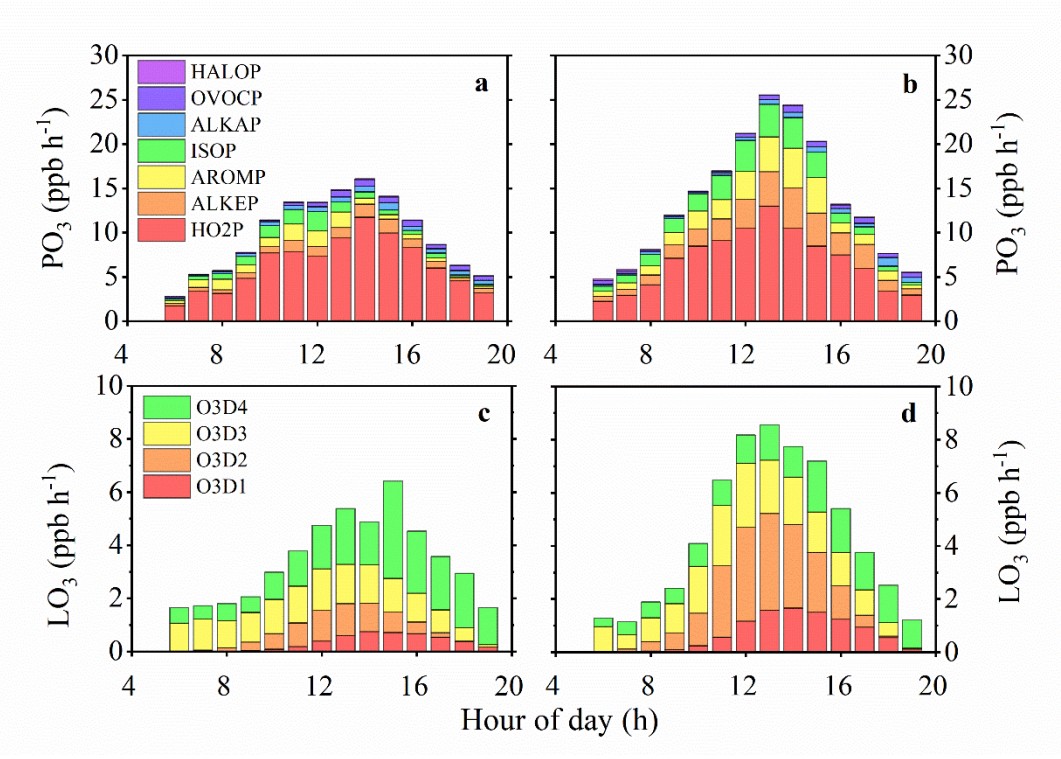

**Figure 3.** Mean diurnal profile of the instantaneous ozone production and destruction rate

calculated from the MCM-OBM model (instantaneous ozone rate derived from observed VOCs

in a and c and from PIC-VOCs in b and d). The upper panel presents the speciation of the ozone

formation rate. The lower panel presents the speciation of the ozone destruction rate. The data

range is August 1-28, 2019.

The budget of OH-HO$_2$-RO$_2$ radicals was further analysed to understand the

photochemical O$_3$ formation process. The comparison of the radical budget derived from the

observed and PIC-VOCs is shown in Figure 4. The radical cycles are divided into radical

sources (green boxes), radical sinks (black boxes), radical propagations (red circles) and

equilibria between radical and reservoir species (yellow boxes). The numbers or percentages

are the average formation rates (ppb h$^{-1}$) or relative contributions of the corresponding reaction

path based on the observed VOCs (outside the brackets) and the PIC-VOCs (inside the brackets)

to a certain radical. The relative contributions of different radical paths based on the observed
VOCs (outside the brackets) were comparable with those reported in Beijing, Shanghai, and
Guangzhou (Tan et al., 2019), while variations were observed for some reaction paths based
on the PIC-VOCs. For example, the reaction between ozone and alkenes based on initial VOC
concentrations (percentages inside the brackets) contributed more to OH (from 7% to 21%)
and $HO_2$ radical production (from 6% to 12%), while photolysis of HONO and HCHO
contributed less to the production of OH (from 76% to 60%) and $HO_2$ radicals (from 44% to
40%), respectively. Other radical sources were consistent between the two scenarios.
Interestingly, the average formation rates of OH, $HO_2$ and $RO_2$ radicals derived from the PIC-
VOCs were obviously higher than those from the observed VOCs. In particular, the oxidation
of NO by $RO_2$ and $HO_2$ increased by 1.6 and 1.3 ppb $h^{-1}$, respectively. The enhanced oxidation
rate of NO was equal to the increase in the average $F(O_3)$ in the analysis process above. This
meant that the radical propagation of OH-$RO_2$-$HO_2$ sped up in the case of PIC-VOCs,
subsequently accelerating the chemical loop of NO-$NO_2$-$O_3$. For the radical sinks and equilibria
related to $HNO_4$, $RONO_2$ and PAN, the values were basically comparable between the two
scenarios. In addition, the $O_3$ formation from the $RO_2$ path increased by 4.1% (from 39.5% to
43.6%) in the simulation using the PIC-VOCs compared with the observed VOCs. The above
budget analysis explained the observed increases in $F(O_3)$ (~3 ppb $h^{-1}$), which were mainly
driven by the reaction of missed reactive VOCs, such as alkenes, with $O_3$.

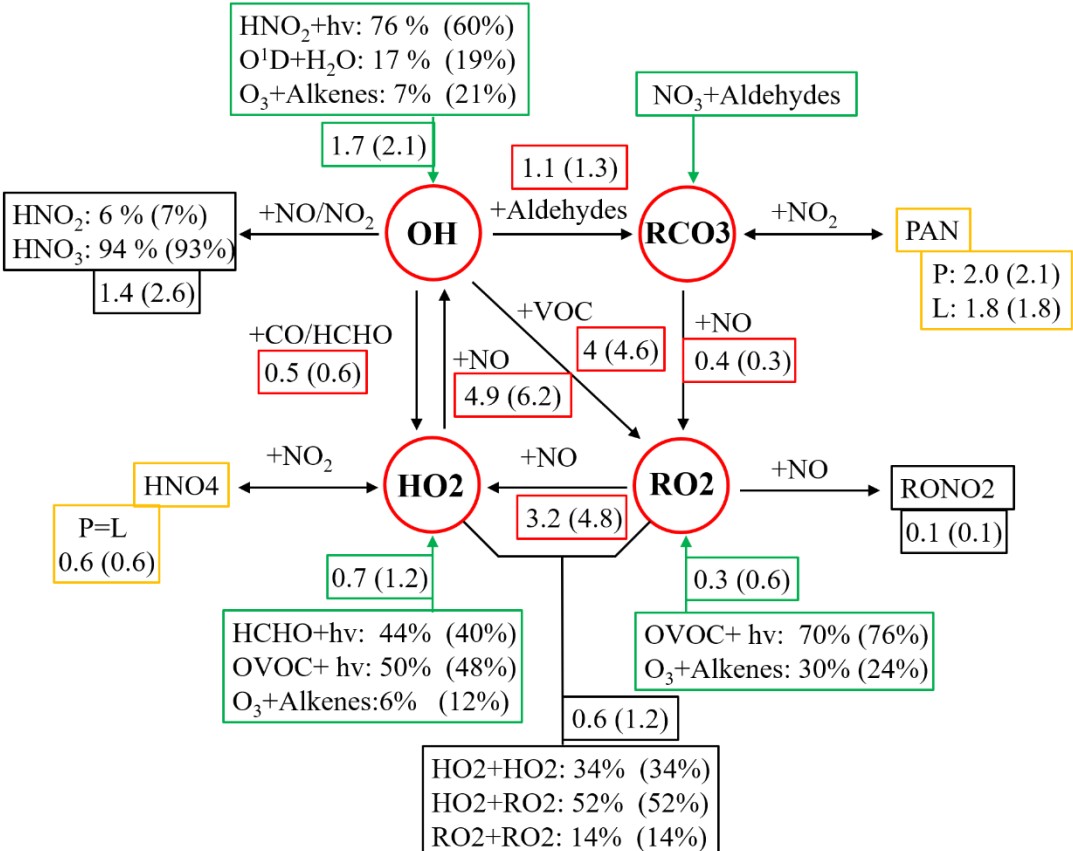


**Figure 4.** Comparison of the OH-HO$_2$-RO$_2$ radical budget derived from the observed and PIC-VOCs under daytime conditions (07:00 to 19:00 LT). The green, black, red and yellow boxes denote the sources of radicals, radical sinks, radical propagation, and racial equilibrium, respectively. The numbers or percentages outside and inside the brackets are the average formation rates (ppb h$^{-1}$) or relative contributions to a specific radical of the corresponding reaction path based on observed VOCs and PIC-VOCs, respectively.

### 3.4 In situ O$_3$ formation process

In addition to chemical processes, which can be simulated using the OBM-MCM model, transport processes, including horizontal, vertical transportation and dry deposition processes (Tan et al., 2019), also have an important influence on the O$_3$ concentration. Thus, the change in instantaneous ozone concentration can reflect the combined effect between photochemical

and physical transport processes (Tan et al., 2019). This change can be expressed as,

$$\frac{dO_x}{dt} = F(O_3) + R(O_3) \quad (6)$$

where $dOx/dt$ is the $O_3$ concentration change rate based on the measured data (ppb h$^{-1}$); $F(O_3)$
is the net $O_3$ formation rate (ppb h$^{-1}$), and $R(O_3)$ indicates transportation (ppb h$^{-1}$). A positive
value of $R(O_3)$ indicates an inflow of $O_3$ with airmass and vice versa. $O_3$ was replaced with $Ox$
($O_3+NO_2$) to correct the titration of $O_3$ by NO (Pan et al., 2015).

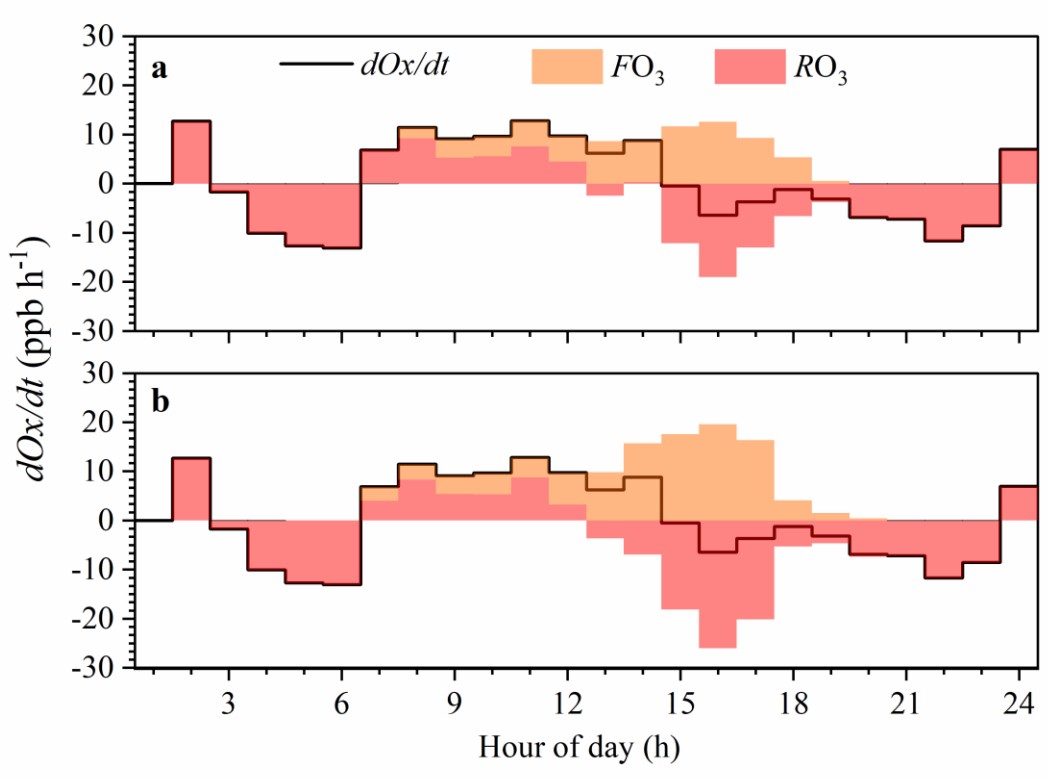


**Fig. 5.** The variation in $Ox$ concentration and formation rate during an $O_3$ pollution episode
(Aug. 1$^{st}$). (a and b present the local ozone formation processes of the measured and PIC-VOCs,
respectively.)
The $O_3$ budget analysis was performed during an $O_3$ pollution episode (Aug. 1$^{st}$). Figure
5 shows the simulated local ozone formation process based on the measured and PIC-VOCs.
The hourly variation in $O_3$ concentrations from 19:00 to 6:00 the next day was dominated by
regional transportation without $O_3$ formation, while local photochemical $O_3$ formation could
explain all or part of the $O_3$ concentration change during the time window from 07:00 to 19:00.
The $d(O_3)/dt$ shows an increase from 07:00 to 15:00 LT. However, $d(O_3)/dt$ sharply changed to
negative values at 16:00, which was consistent with diurnal $O_3$ (the $O_3$ peaks at 15:00) in Figure

428   1.

The average daytime $F(O_3)$ based on the observed and photochemical initial
concentrations was $6.4\pm4.0$ and $8.9\pm6.7$ ppb $h^{-1}$, respectively. Photochemical $O_3$ formation
under both conditions started at 07:00 and reached maximum values of 12.6 and 19.6 ppb $h^{-1}$
at 15:00, respectively. The maximum daily value of $P(O_3)$ was higher than those in the urban
areas of Japan, America, and England (Whalley et al., 2018; Ren et al., 2006; Griffith et al.,
2016; Kanaya et al., 2009) and lower than those in the suburbs of Guangzhou (Lu et al., 2012)
and the urban areas and suburbs of Beijing (Lu et al., 2013). Before 12:00, the $O_3$ formation
rate based on the PIC-VOCs was slightly higher than that based on the measured VOCs, while
both rates were within a range of $2.0 \sim 6.5$ ppb $h^{-1}$. From 12:00 to 17:00, the $O_3$ formation rate
based on the PIC-VOCs and the observed concentration of VOCs greatly increased due to
active photochemistry.
As shown in Figure 5, the increased $O_3$ concentration was larger than the local $O_3$
photochemical production from 07:00 to 12:00 ($R(O_3)$ was positive). This was mainly because,
under stable conditions, the nighttime residual layer (RL) is isolated from mixing with the
nighttime surface layer (Tan et al., 2021). The RL layer usually contains an air mass with a
higher ozone mixing ratio than in the surface layer. In the morning, surface heating causes
mixing upward in the surface layer until the temperature inversion is eroded away and rapid
mixing of pollutants throughout the surface and boundary layer occurs. However, $R(O_3)$ was
negative in the afternoon, which indicated that the local $O_3$ formation at the measurement site
contributed to not only the changes in the in situ $O_3$ concentration but also the $O_3$ source of the
downwind regions. This was more clearly shown in Figure 4B under the PIC-VOCs condition.
These results illustrated that local $O_3$ photochemistry played a crucial role in both the local and
regional $O_3$ concentrations, which can be underestimated if consumed VOCs with high
reactivities are ignored.

## 4. Conclusions

In this study, we presented the local $O_3$ formation process in August 2019 in Beijing based
on the concentrations of observed and PIC-VOCs. The mean diurnal profile of $O_3$ was
unimodal with a peak at 15:00, while $NOx$ and observed TVOCs showed an opposite diurnal
curve, and the PICs of TVOCs showed a different diurnal curve compared with that of the
observed VOCs, with a slight increase from 07:00 to 14:00. The EKMA curve indicated that
instantaneous $O_3$ production was dependent on the real-time concentrations of $NOx$ and VOCs,
i.e., the VOC-limited regime in the morning (09:00-10:00) and the $NOx$-limited regime at noon
(14:00-15:00). The sensitivity regime of $O_3$ formation could be misdiagnosed if the consumed
VOCs are not considered, for example, the VOC-limited regime (observed) shift to a transition
regime (PIC-VOCs) in the morning is ignored. The mean $F(O_3)$ based on PIC-VOCs was 3.0
ppb $h^{-1}$ higher than that based on the measured VOCs, indicating that the underestimation of
local photochemistry in the local $O_3$ concentration could reach ~36 ppb day$^{-1}$ if the consumed
VOCs are not accounted for. And the mean ozone concentration of downwind site was 27.6
ppb day$^{-1}$ higher than the observation site, slightly lower than the difference (~36 ppb day$^{-1}$)
between PIC-VOCs and observed VOCs, which indirectly supported the accuracy of the above
results. The radical budget analysis explained the observed increases in $F(O_3)$ (3 ppb h$^{-1}$), which
were mainly driven by the reaction of missed reactive VOCs, such as alkenes, with $O_3$. In
addition, the OH-HO$_2$ radical cycle was obviously accelerated by highly reactive alkenes after
the photochemical loss of VOCs was accounted for. Finally, the results of the in situ $O_3$
formation process indicated that local $O_3$ photochemical formation played a key role in both
local and regional $O_3$ concentrations. In conclusion, our results suggested that PIC-VOCs were
more suitable than the observed VOC concentrations for diagnosing $O_3$ formation sensitivity.

**Author contributions:** Wei Ma: Methodology, data curation, and writing of the original draft;
Zemin Feng: Methodology, investigation, data curation, and writing of the original draft; Junlei
Zhan: Methodology, investigation, data curation; Yongchun Liu: Conceptualization,
investigation, data curation, writing, reviewing & editing, supervision, and funding acquisition;
Pengfei Liu: Methodology, investigation, data curation, writing, reviewing & editing;
Chengtang Liu: Methodology, investigation, data curation, writing, reviewing & editing;
Qingxin Ma: Methodology, investigation, data curation; Kang Yang: Methodology,
investigation, data curation. Yafei Wang: Methodology, investigation, resources, data curation;
Hong He: Resources, writing, reviewing & editing; Markku Kulmala: Methodology, writing,
reviewing & editing; Yujing Mu: Conceptualization, methodology, data curation, writing,
reviewing & editing. Junfeng Liu: Conceptualization, methodology, data curation, writing,
reviewing & editing, and supervision.
**Competing interests:** The authors declare that they have no substantive conflicts of interest.
**Data availability:** Data are available upon request to Yongchun Liu (liuyc@buct.edu.cn).
**Acknowledgements:** This research was financially supported by the National Natural Science
Foundation of China (41877306, 92044301, 21976190), the Ministry of Science and
Technology of the People's Republic of China (2019YFC0214701), the Strategic Priority
Research Program of the Chinese Academy of Sciences and Beijing University of Chemical
Technology.

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
