# Peer review of "Influence of Photochemical Loss of VOCs on Understanding Ozone"

_Atmospheric Chemistry and Physics, 2021_

## Referee Comment (RC1)

**Review of "Influence of Photochemical Loss of VOCs on Understanding Ozone Formation Mechanism"**

November 24, 2021

Ma et al. "Influence of Photochemical Loss of VOCs on Understanding Ozone Formation Mechanism" uses measurements in Beijing to demonstrate how failing to account for oxidation of VOCs between the point of emission and measurement can lead to misinterpretation of the dominant chemical regime for ozone production and a misestimate of the rate of ozone production. The authors use measurements of xylene and ethylbenzene to compute the OH exposure since time of emission, assuming that the concentrations of these species between 5 and 6 AM are a good estimate for their concentrations with zero oxidation. They then use that OH exposure along with rate constants for reaction of other VOCs with OH to back out the "photochemical initial concentrations" (PICs) of VOCs. They compare the net ozone production between the measured and PIC VOC concentrations using a box model with the Master Chemical Mechanism and use another box model with the RACM2 mechanism to examine the dominant ozone production chemical regime with these different VOC concentrations. They conclude that using the measured VOC concentrations underestimates the ozone production rates.

The argument that one must consider the effect of photochemical oxidation of VOCs between source and measurement to accurately determine the best approach to reduce ozone exposure is an important one; however, I do not see that this paper adds much to our understanding of this issue, either generally or specific to Beijing. Additionally, elements of the methodology require further explanation or justification. I will discuss these factors in detail below. Taken together, this paper should be reconsidered after major revision.

**Major comments**

**Novelty**

My largest concern is that it is not clear what this paper adds to our understanding in regards to the effect of VOC degradation on estimated $O_3$ production rates. This question

has previous been addressed by several papers cited in the introduction to this one (e.g. Shao et al., 2011; Gao et al., 2018) as well as others (e.g. Xie et al., 2008; Shao et al., 2009; Li et al., 2015). Near the end of the introduction, this paper states that "it is unclear how the highly reactive VOCs, which [have] degraded during transport from the source to the receptor site, will affect the instantaneous formation process of $O_3$." Yet Li et al. (2015) addresses this in Sect. 3.3. of their paper:

> VOC species may experience photochemical losses from emission sources to receptor sites, which are important for ground-level ozone formation. It is very likely that the OFP [ozone formation potential] is underestimated when using only the observed mixing ratio of ambient VOCs at a receptor site. Therefore, the initial mixing ratios of VOCs must be considered if ozone abatement measures are to be implemented.... We calculated the OFP for the initial VOCs (except carbonyls) to avoid misjudging the role of the major VOC species in ozone formation. The OFPs calculated based on the initial mixing ratio of VOC species differed from those based on observations. The OFPs for total NMHCs (excluding isoprene) increased by 16.09% (from 59.60 to 69.18 ppbv O3), 12.06% (from 33.46 to 37.50 ppbv O3), and 3.38% (from 68.89 to 71.22 ppbv O3) after correction for chemical conversion at GC, QZ, and BJ, respectively.... In this region, the OFPs for trans-2-butene, cis-2-butene, cis-2-pentene, and isoprene would be underestimated by up to 40% if chemical losses were neglected.

This analysis by Li et al. (2015) previously showed that not accounting for chemical loss of VOCs will underestimate $O_3$ production. Replication and confirmation are valuable, but the authors must do a better job placing their paper in the context of earlier work and, especially if replication is the focus, compare their results to previous studies.

**Methlogical concerns**

There are several elements of the methodology that I have concerns about. I will order this section from most to least severe.

**Choice of initial xylene and ethylbenzene:** A key part of this methodology is the use of xylene and ethylbenzene as a chemical clock to compute the integrated OH exposure for all VOCs from emission to measurement. This requires knowing the initial concentrations of xylene and ethylbenzene; for the purposes of this paper, the concentrations between 5 and 6 AM are considered the initial values. This is presumably the last hour before sunrise (and so the last measurement before OH chemistry initiates), but I did not see where the rationale for this selection is given in the paper. The reasoning for that selection should be made clear.

My larger issue with this approach is that it implicitly assumes that the source of xylene and ethylbenzene remains constant throughout the day. This is a risky assumption: shifts in the

wind direction or changes in upwind emissions could alter the source emission ratio of xylene to ethylbenzene throughout the day. I did not see anywhere in the paper where the authors carried out a back trajectory or other source area analysis to determine if the assumption of consistent xylene and ethylbenzene sources throughout the day is correct. Without that analysis, we cannot know if the 5-6 AM xylene and ethylbenzene concentrations are a reasonable approximation of the initial concentrations for all airmasses measured throughout the day.

**Validation of adjusted $O_3$ production:** In the conclusion, the authors claim that, "The radical budget analysis illustrated that the $O_3$ formation processes between the observed and photochemical initial VOCs showed no significant difference, **but the former one underestimated the $O_3$ production rate obviously**" (emphasis added). While I agree in principle that using the observed VOCs underestimates the $O_3$ production, it is unclear how well the method presented in this paper corrects for that underestimation, as I saw no comparison against any truth metric for $O_3$ production.

One method to check the accuracy of the authors' approach would be to use a pair of measurement sites, one upwind and one downwind, with the upwind site measuring $O_3$ concentration. Combined with a trajectory analysis, one could potentially compute the $O_3$ production based on the difference in concentrations between the two sites (though mixing may complicate this), and compare that to the modeled $O_3$ production using observed and PIC VOCs. If an upwind $O_3$ site is not available, an analysis using $dO_x/dt$ as in Fig. 5, except with independent constraint on the transport of $O_3$ (perhaps from a regional model or satellite observations) may be another option.

**VOC correction:** In Fig. 1, the difference between the observed and "photochemical initial concentration" (PIC) VOCs is zero before 6a and after 7p (19:00). Between 6a and 7p, the offset between the observed and PIC VOC concentrations seems almost (but not quite) constant. What seems particularly odd is how the observed-PIC difference jumps from nothing to essentially its maximum value between 5a and 6a, then likewise drops instantaneously from its maximum value to zero between 7p and 8p. I would expect the transition to be more gradual, with photolysis (and therefore OH concentrations) being less in the hour immediately following sunrise than later in the morning (and vice versa at night). It would be helpful if the authors provided a timeseries (at sub-hourly resolution) of the concentration of xylene and ethylbenzene, their ratio, the OH exposure derived from these quantities, and the solar zenith angle, to demonstrate how the OH exposure correction changes with time of day.

**Ozone production and loss metrics:** Please discuss for Eq. 3 how alkyl nitrate formation is treated; is $k_{RO_2+NO}$ the rate constants for only $RO_2 + NO$ reactions that produce RO and $NO_2$? If $k_{RO_2+NO}$ is the rate for all $RO_2 + NO$ reactions, then the alkyl nitrate branching ratio must be accounted for.

Relately, in Fig. 3, a comparison of panels (c) and (d) appears to indicate that the loss of ozone via reaction of $O^{1D}$ with $H_2O$ increases when using PIC VOCs rather than measured.

Please elaborate why this is, is this just because there is more $O_3$ (and so more $O^{1D}$) in the model with PIC VOCs, and so the rate increases even through the amount of $H_2O$ remains constant? If so, it might help to include a companion figure to Fig. 3 in the supplement that shows $P(O_3)$ and $L(O_3)$ as percentages of $O_3$ production/loss to help the reader understand the relative change in loss processes as well.

**Minor comments**

- The argument made in lines 66–71 of the introduction about the different mixing ratios of VOCs at the source vs. measurement site is confusing on a first read because it is not clear that the scenario which applies here is where the source one is attempting to control with policy is significantly upwind of the measurement site. If we were considering a source (e.g. traffic) which is closely clustered around the measurement site, then the VOCs measured at the site will be the correct concentrations to consider for $O_3$ production.

- It is unclear whether Fig. 1 and (to a lesser extent) Fig. 3 are for one specific day or the entire campaign. For these figures, please specify the time period considered (since Fig. 5 is specific to one day).

- For Fig. 1, specify which series are measurements and which are calculated. (I assume all but the TVOC_PIC series are measurements, but please be explicit.)

- For Fig. 2, define specific times (i.e. "8 AM" or "11 AM to 1 PM") rather than "morning" and "noon" so that we can compare to Fig. 1. Also, line 236 seems to imply that the "noon" points are actually 15:00? That is confusing.

- Please explain in the caption what the percentages in Fig. 4 represent; I only saw a description of the other numbers as production rates in ppb $h^{-1}$. In general, the discussion of the radical chain on lines 296–309 is pretty dense and difficult to follow, but I cannot give any suggestions to improve it without understanding what all the elements in Fig. 4 are.

**Summary**

While this paper is a fair study of how one might account for degradation of VOCs between sources and measurements in order to formulate better approaches to controlling $O_3$ production, there have been a number of earlier studies looking at this problem in Beijing. In my opinion, in order for a revision to be considered for publication, the authors must revise the paper to clarify what new information their work adds compared to the previous studies *or* refocus the paper as a replication study or an update to more recent times. In this second case, the revision should include a thorough comparison with previous studies of this effect in the Beijing area.

**References**

Gao, J., Zhang, J., Li, H., Li, L., Xu, L., Zhang, Y., Wang, Z., Wang, X., Zhang, W., Chen, Y., Cheng, X., Zhang, H., Peng, L., Chai, F., and Wei, Y.: Comparative study of volatile organic compounds in ambient air using observed mixing ratios and initial mixing ratios taking chemical loss into account – A case study in a typical urban area in Beijing, Science of The Total Environment, 628-629, 791–804, https://doi.org/10.1016/j.scitotenv.2018.01.175, 2018.

Li, L., Xie, S., Zeng, L., Wu, R., and Li, J.: Characteristics of volatile organic compounds and their role in ground-level ozone formation in the Beijing-Tianjin-Hebei region, China, Atmospheric Environment, 113, 247–254, https://doi.org/10.1016/j.atmosenv.2015.05.021, 2015.

Shao, M., Lu, S., Liu, Y., Xie, X., Chang, C., Huang, S., and Chen, Z.: Volatile organic compounds measured in summer in Beijing and their role in ground-level ozone formation, Journal of Geophysical Research: Atmospheres, 114, https://doi.org/10.1029/2008JD010863, 2009.

Shao, M., Wang, B., Lu, S., Yuan, B., and Wang, M.: Effects of Beijing Olympics Control Measures on Reducing Reactive Hydrocarbon Species, Environ. Sci. Technol., 45, 514–519, https://doi.org/10.1021/es102357t, 2011.

Xie, X., Shao, M., Liu, Y., Lu, S., Chang, C.-C., and Chen, Z.-M.: Estimate of initial isoprene contribution to ozone formation potential in Beijing, China, Atmospheric Environment, 42, 6000–6010, https://doi.org/10.1016/j.atmosenv.2008.03.035, 2008.

---

## Author Comment (AC1)

Dear Reviewer,

We appreciate your careful consideration of our manuscript. We have carefully responded to all of your **point-by-point** comments and issues and have revised the manuscript accordingly. These revisions are described in detail below.

Ma et al. "Influence of Photochemical Loss of VOCs on Understanding Ozone Formation Mechanism" uses measurements in Beijing to demonstrate how failing to account for oxidation of VOCs between the point of emission and measurement can lead to misinterpretation of the dominant chemical regime for ozone production and a misestimate of the rate of ozone production. The authors use measurements of xylene and ethylbenzene to compute the OH exposure since time of emission, assuming that the concentrations of these species between 5 and 6 AM are a good estimate for their concentrations with zero oxidation. They then use that OH exposure along with rate constants for reaction of other VOCs with OH to back out the "photochemical initial concentrations" (PICs) of VOCs. They compare the net ozone production between the measured and PIC VOC concentrations using a box model with the Master Chemical Mechanism and use another box model with the RACM2 mechanism to examine the dominant ozone production chemical regime with these different VOC concentrations. They conclude that using the measured VOC concentrations underestimates the ozone production rates.

The argument that one must consider the effect of photochemical oxidation of VOCs between source and measurement to accurately determine the best approach to reduce ozone exposure is an important one; however, I do not see that this paper adds much to our understanding of this issue, either generally or specific to Beijing. Additionally, elements of the methodology require further explanation or justification. I will discuss these factors in detail below. Taken together, this paper should be reconsidered after major revision.

   **Response:** Thank you for your good comments and suggestions. We will reply your concerns point-by-point below.

**Major comments**

1. My largest concern is that it is not clear what this paper adds to our understanding in regards to the effect of VOC degradation on estimated $O_3$ production rates. This question has previous been addressed by several papers cited in the introduction to this one (e.g. Shao et al., 2011; Gao et al., 2018) as well as others (e.g. Xie et al., 2008; Shao et al., 2009; Li et al., 2015). Near the end of the introduction, this paper states that "it is unclear how the highly reactive VOCs, which [have] degraded during transport from the source to the receptor site, will affect the instantaneous formation process of $O_3$." Yet Li et al. (2015) addresses this in Sect. 3.3. of their paper:

> "VOC species may experience photochemical losses from emission sources to receptor sites, which are important for ground-level ozone formation. It is very likely that the OFP [ozone formation potential] is underestimated when using only the observed mixing ratio of ambient VOCs at a receptor site. Therefore, the initial mixing ratios of VOCs must be considered if ozone abatement measures are to be implemented.... We calculated the OFP for the initial VOCs (except carbonyls) to avoid misjudging the role of the major VOC species in ozone formation. The OFPs calculated based on the initial mixing ratio of VOC species differed from those based on observations. The OFPs for total NMHCs (excluding isoprene) increased by 16.09% (from 59.60 to 69.18 ppbv $O_3$), 12.06% (from 33.46 to 37.50 ppbv $O_3$), and 3.38% (from 68.89 to 71.22 ppbv $O_3$) after correction for chemical conversion at GC, QZ, and BJ, respectively.... In this region, the OFPs for trans- 2-butene, cis-2-butene, cis-2-pentene, and isoprene would be underestimated by up to 40% if chemical losses were neglected."

This analysis by Li et al. (2015) previously showed that not accounting for chemical loss of VOCs will underestimate $O_3$ production. Replication and confirmation are valuable, but the authors must do a better job placing their paper in the context of earlier work and, especially if replication is the focus, compare their results to previous studies.

**Response:** Thank you for your comments and suggestion. As you mentioned, the influence of photochemical loss of VOCs on OFPs estimation had been discussed based

on the maximum incremental reactivities (MIR) in previous work. However, the application of such calculations using the MIR is restricted to areas or episodes in which the $O_3$ formation is VOC-sensitive (Carter, 1994). In the troposphere, the sensitivity of ozone formation on NO$x$ and VOCs varies greatly. Thus, the non-linear relationship between ozone and VOCs/NO$x$ cannot be well elaborated using the MIR method, and a quantitative analysis is necessary for explicitly understanding ozone formation process and mechanisms in the real atmosphere.

In this work, we carried out a thorough analysis on ozone formation using a box model after considering the photochemical loss of VOCs under more realistic atmospheric conditions compared with the MIR method. Our results demonstrated that the ozone sensitivity could be misdiagnosed if one not considering the photochemical loss of VOCs. The contribution of different precursors varied obviously using their initial VOCs concentrations when compared with the observed values, in particular, the contributions of highly reactive alkenes to the $RO_2$ formation were obviously underestimated using the observed VOCs. In addition, the OH-HO$_2$ radical cycle was obviously accelerated by the highly reactive alkenes after photochemical loss of VOCs was accounted for. Although this is generally consistent with these previous studies, we discussed this issue based on quantitative analysis including the instantaneous $O_3$ production rates and the budget of the crucial radicals with the initial concentrations of different precursors. This would provide a technical guidance for regional ozone pollution prevention.

In order to clarify the novelty of this work, we have added more details based on a thorough review of previous work "In evaluating the importance of initial VOCs to ozone production, Xie et al. (2008) found that the OFP at a Peking University site increased by 70% after accounting for the photochemical loss of VOCs. Li et al. (2015) also showed that the OFPs of total NMHCs (excluding isoprene) increased by 16.1% (from 59.6 to 69.2 ppb $O_3$), 12.1% (from 33.5 to 37.5 ppb $O_3$), and 3.4% (from 68.9 to

71.2 ppb $O_3$) after correcting for photochemical loss in Gucheng, Quzhou, and Beijing, respectively. Gao et al. (2018) reported that the OFP could be underestimated by 23.4% (62.4 ppb $O_3$) in Beijing if the photochemical loss of VOCs is not considered. Zhan et al. (2021) found that based on measured VOCs, the OFP increased from 57.8 ppb to 103.9 ppb using the initial VOCs. All the previous work was based on the maximum incremental reactivities (MIR) method. However, the application of such calculations using the MIR method is restricted to areas or episodes in which $O_3$ formation is VOC-sensitive (Carter, 1994). In the troposphere, the sensitivity of ozone formation to $NOx$ and VOCs varies greatly, as evidenced by the wide range of OFP underestimations from ~3% to 70% in previous work. Thus, the nonlinear relationship between ozone and VOCs/$NOx$ cannot be well described using the MIR method, and a quantitative analysis is necessary to explicitly understand the ozone formation process and its mechanisms in the atmosphere." in lines 78-94 in the revised manuscript.

Methodological concerns

2. There are several elements of the methodology that I have concerns about. I will order this section from most to least severe. Choice of initial xylene and ethylbenzene: A key part of this methodology is the use of xylene and ethylbenzene as a chemical clock to compute the integrated OH exposure for all VOCs from emission to measurement. This requires knowing the initial concentrations of xylene and ethylbenzene; for the purposes of this paper, the concentrations between 5 and 6 AM are considered the initial values. This is presumably the last hour before sunrise (and so the last measurement before OH chemistry initiates), but I did not see where the rationale for this selection is given in the paper. The reasoning for that selection should be made clear.

**Response:** Thank you for your good suggestion. Firstly, we choose the initial concentrations of xylene and ethylbenzene based on the diurnal variation of solar irradiation during our observations. As shown in Figure R1, the $J_{NO2}$ increased at 7:00 AM, and in order to eliminate the influence of photochemical process, the concentrations at 5:00 and 6:00 AM were set as the initial concentration. In the revised SI, we have added this Figure as Figure S3, and updated the sentence "we chose the mean concentrations of xylene and ethylbenzene at 05:00-06:00 as their initial concentrations before sunrise according to the ambient $J_{NO2}$ (Figure S3) to calculate the photochemical loss of OH exposure" in lines 157-160 in the revised manuscript.

[Figure]

**Figure R1.** The mean diurnal curve of $J_{NO2}$.

3. My larger issue with this approach is that it implicitly assumes that the source of xylene and ethylbenzene remains constant throughout the day. This is a risky assumption: shifts in the wind direction or changes in upwind emissions could alter the source emission ratio of xylene to ethylbenzene throughout the day. I did not see anywhere in the paper where the authors carried out a back trajectory or other source area analysis to determine if the assumption of consistent xylene and ethylbenzene sources throughout the day is correct. Without that analysis, we cannot know if the 5-6 AM xylene and ethylbenzene concentrations are a reasonable approximation of the initial concentrations for all airmasses measured throughout the day.

**Response:** Thank you for your good suggestions. In previous work (Shao et al., 2011; Zhan et al., 2021), it has been justified for selecting the pair of ethylbenzene/xylene as the tracers when one calculating ambient OH exposure in terms of the following rules: 1) the concentrations of xylene and ethylbenzene are well correlated, which indicates that they are simultaneously emitted (Figure R2 and Figure S4); 2) they have different degradation rates in the atmosphere; 3) the calculated PICs are in good agreement with those calculated using other tracers, such as *i*-butene/propene (Figure R3).

[Figure]

**Figure R2.** The relationship between the concentration of ethylbenzene and xylene.

[Figure]

**Figure R3.** Comparison of PICs calculated for xylene/ethylbenzene and i-

Butene/Propene. (Error bars are standard deviations.)

In addition, the stability of the emission rates can be evaluated with the ambient ratio for a specific pair of VOCs with similar degradation rate constants (Golden et al., 2000; Jobson et al., 2004), such as benzene *vs* acetylene, trans-2-butene *vs* cis-2-butene, ethene *vs* toluene, and n-hexane *vs* toluene (Shao et al., 2011). Figure R4 (same as Figure S5) shows the correlation between benzene and acetylene, and between n-hexane and toluene during our observations. The linear correlation coefficients ($R^2$) were generally higher than 0.7, which was close to that reported by Shao et al (2011). This means that the emissions of the primary hydrocarbons are relatively constant throughout the day.

[Figure]

**Figure R4.** The relationship between the concentration of benzene vs acetylene and n-hexane vs toluene.

To further check the assumption that the emissions of xylene and ethylbenzene were constant throughout the day, their potential sources have been calculated using a source-receptor model (the potential source contribution function, PSCF) during our observations. As shown in Figure R5 (same as Figure S6), besides the similar trajectories at 5:00 and 6:00 and during the daytime, xylene and ethylbenzene showed the similar distribution. In addition, the ratio of ethylbenzene/xylene at 5:00 and 6:00 were similar to that during the daytime. This means that the wind field was relatively stable during our observations and the emissions of xylene and ethylbenzene were constant throughout the day.

[Figure]

**Figure R5.** The potential source contribution function (PSCF) maps for the ratio of xylene to ethylbenzene (a and b), ethylbenzene (c and d), and xylene (e and f) arriving in the observation site. The figures of a, c and e are the results of 05:00 and 06:00, and the figures of b, d and f are the results during the daytime (07:00-19:00).

In the revised manuscript, we added the sentences "In previous work (Shao et al., 2011; Zhan et al., 2021), the selection of ethylbenzene and xylene as tracers was justified for calculating ambient OH exposure under the following conditions: 1) the concentrations of xylene and ethylbenzene were well correlated (Figure S4), which indicated that they were simultaneously emitted; 2) they had different degradation rates in the atmosphere; and 3) the calculated PICs were in good agreement with those calculated using other tracers, such as i-butene/propene (Zhan et al., 2021). To test the relative constant emission ratio from different sources, we chose benzene vs. acetylene

and n-hexane vs. toluene as references, and the result is shown in Figure S5. These ambient ratios could directly reflect their relative emission rates from sources (Goldan et al., 2000; Jobson et al., 2004). The linear correlation coefficients ($R^2$) were generally higher than 0.7, which were equal to that reported by Shao et al. (2011). To further test the assumption that the emissions of xylene and ethylbenzene were constant throughout the day, their potential sources were calculated using a source-receptor model (the potential source contribution function, PSCF). As shown in Figure S6, xylene and ethylbenzene showed similar distributions. In addition, the ratio of ethylbenzene/xylene at 5:00 and 6:00 was similar to that during the daytime. These results indicated that the emissions of xylene and ethylbenzene were constant throughout the day." in lines 160-175.

4. Validation of adjusted $O_3$ production: In the conclusion, the authors claim that, "The radical budget analysis illustrated that the $O_3$ formation processes between the observed and photochemical initial VOCs showed no significant difference, but the former one underestimated the $O_3$ production rate obviously" (emphasis added). While I agree in principle that using the observed VOCs underestimates the $O_3$ production, it is unclear how well the method presented in this paper corrects for that underestimation, as I saw no comparison against any truth metric for $O_3$ production.

**Response:** Thank you for your comments. We agree with you that it is better to compare the $O_3$ production with the truth metric for $O_3$ production. However, it is impossible to directly measure the truth metric for $O_3$ production in the atmosphere at the present time, subsequently, to answer the question how well the method presented here corrects for that underestimation. On the other hand, we had to constrain the ozone concentrations when simulating the ozone formation process (Lu et al., 2013; Tan et al., 2017). Thus, it was also impossible to directly compare the ozone production between the two scenarios. We compared the net ozone production rates other than ozone production or concentrations in the two scenarios and further compared the ozone production by integrating the net formation rates in the day (7:00-19:00). The calculated ozone production based on the initial VOCs concentrations was ~36 ppb day$^{-1}$ higher

than that using the measured VOCs concentrations. We think this is reasonable and the best way we can do at the present time. In the revised manuscript, we made it clearer as It should be pointed out that it is better to compare $O_3$ production with the true metric for $O_3$ production. However, it is impossible to directly measure the true metric for $O_3$ production in the atmosphere at the present time to know how well the method presented here corrects for that underestimation. In addition, the ozone concentrations must be constrained when simulating the ozone formation process (Lu et al., 2013; Tan et al., 2017). Thus, it is impossible to directly compare the ozone production based on PIC-VOCs with that using measured VOCs concentrations. Therefore, we alternatively compared the integrated net ozone production rates rather than ozone production or concentrations between the two scenarios" in lines 316-324.

5. One method to check the accuracy of the authors' approach would be to use a pair of measurement sites, one upwind and one downwind, with the upwind site measuring $O_3$ concentration. Combined with a trajectory analysis, one could potentially compute the $O_3$ production based on the difference in concentrations between the two sites (though mixing may complicate this), and compare that to the modeled $O_3$ production using observed and PIC VOCs. If an upwind $O_3$ site is not available, an analysis using dOx/dt as in Figure 5, except with independent constraint on the transport of $O_3$ (perhaps from a regional model or satellite observations) may be another option.

**Response:** Thank you for your good suggestion. We absolutely agree with you that an upwind ozone and VOCs measurement combining with a trajectory analysis might provide an approach to check the accuracy of our results. Alternatively, a transient ozone production rate analysis after subtracting the transport of ozone with a regional model and/or satellite observation may be another option. Unfortunately, neither the upwind measurement nor the regional model simulation was available at the present time. But according to trajectory analysis, we choose 4th August as the case to

explore the influence of the transport of ozone on one downwind site. As showed in Figure R6 (same as Figure S12), the mean ozone concentration of the downwind site (national monitoring station, NMS) was $27.6 \pm 21.8$ ppb day$^{-1}$ higher than the observation site (OS), slightly lower than the difference (~36 ppb day$^{-1}$) between PIC-VOCs and observed VOCs, which indirectly supported the accuracy of our results. We added the corresponding analysis in the revised manuscript in lines 324-334 "An upwind $O_3$ and VOCs measurement combined with a trajectory analysis might provide an approach for checking the accuracy of our results. Alternatively, conducting a transient $O_3$ production rate analysis after subtracting the transport of $O_3$ with a regional model and/or satellite observation might be another option. Unfortunately, neither the upwind measurement nor the regional model simulation was available at the time of our study. To further check the accuracy of our results, we chose August 4$^{th}$ as a test case to explore the influence of the transport of ozone on a downwind site based on the trajectory analysis. As shown in Figure S12, the mean ozone concentration of the downwind site (national monitoring station, NMS) was $27.6 \pm 21.9$ ppb day$^{-1}$ higher than that of the observation site (OS), which was slightly less than the difference (~36 ppb day$^{-1}$) between PIC-VOCs and observed VOCs and indirectly rationalized our results.".

[Figure]

**Figure R6.** The wind direction at the observation site on 4$^{th}$ Aug (a, backward and forward meaning upwind and downwind, respectively) and the diurnal variation of ozone concentration at observation site (OS) and one downwind site (national monitoring station (NMS)).

6. VOC correction: In Figure 1, the difference between the observed and "photochemical initial concentration" (PIC) VOCs is zero before 6a and after 7p (19:00). Between 6a and 7p, the offset between the observed and PIC VOC concentrations seems almost (but not quite) constant. What seems particularly odd is how the observed-PIC difference jumps from nothing to essentially its maximum value between 5a and 6a, then likewise drops instantaneously from its maximum value to zero between 7p and 8p. I would expect the transition to be more gradual, with photolysis (and therefore OH concentrations) being less in the hour immediately following sunrise than later in the morning (and vice versa at night). It would be helpful if the authors provided a timeseries (at sub-hourly resolution) of the concentration of xylene and ethylbenzene, their ratio, the OH exposure derived from these quantities, and the solar zenith angle, to demonstrate how the OH exposure correction changes with time of day.

   **Response:** Thank you for your comments and good suggestion. The transition of the initial VOCs should gradually increase with photolysis as you pointed out. Figure R7 (same as FigureS7) shows the mean diurnal curves of the concentrations of xylene and ethylbenzene, their ratio, the OH concentration and the solar zenith angle. The variation of solar zenith angle was similar to that of $J_{NO2}$ (Figure R1). The OH

concentration was from $0.82$ to $8.1 \times 10^6$ $(4.3 \pm 1.9 \times 10^6)$, the ratio of xylene to ethylbenzene increased gradually (07:00~12:00). In the revised manuscript in lines 177-181, we added the sentences "The ratio of xylene to ethylbenzene and the OH exposure concentration are shown in Figure S7. The results showed that the ratio of xylene to ethylbenzene increased gradually (07:00~12:00), which is consistent with the trend of xylene and ethylbenzene. The OH exposure was from $0.82$ to $8.1 \times 10^6$ molecule cm$^{-3}$ h, with a mean daytime value of $4.3 \pm 1.9 \times 10^6$ molecules cm$^{-3}$ h.".

[Figure]

**Figure R7.** The mean diurnal curves of the concentration of xylene and ethylbenzene, their ratio, the OH exposure concentration (molecule cm$^{-3}$), and solar zenith angle.

We double checked the PIC concentrations and the related dataset from 07:00 to 19:00 in Figure 1. We are sorry for a stupid mistake when drawing Figure 1. We wrongly added the data of 05:00 and 06:00 twice when calculating PICs of VOCs from 07:00 to 12:00, thus leading to the sharp increase in the early morning (07:00-10:00). The simulation process using the PIC-VOCs is correct. And we have corrected the Figure 1 (Figure R8) and the updated the sentences in lines 259-265 in the revised manuscript

"However, the photochemical initial concentrations (PICs) of TVOCs, which varied from 2.2 to 27.8 ppb with a mean value of 24.5±2.1 ppb, showed a different diurnal curve compared with the observed concentrations. It slightly increased from 07:00 to 14:00, which was similar to the diurnal variation of VOCs in previous work (Zhan et al., 2021). The average PIC-VOCs was 6.9±0.5 ppb higher than the observed concentration of TVOCs, indicating an underestimated contribution of the local photochemistry of VOCs to $O_3$ and organic aerosol formation."

[Figure]

**Figure R8.** Overview of average diurnal variations of $O_3$, NO$x$, and TVOC. The data represent measured results, except for those of the TVOC_PIC, which are calculated based on OH radical exposure. The data range is August 1-28, 2019.

7. Ozone production and loss metrics: Please discuss for Eq. 3 how alkyl nitrate formation is treated; is $k_{RO2+NO}$ the rate constants for only RO$_2$ + NO reactions that produce RO and NO$_2$? If $k_{RO2+NO}$ is the rate for all RO$_2$ + NO reactions, then the alkyl

nitrate branching ratio must be accounted for.

**Response:** Thank you for your comments. We agree with you that the reaction between $RO_2$ and $NO$ can produce both $RO$ and alkyl nitrate. The branching ratio was actually accounted for in the model. For example,

$3.00 \times 10^{-12} \times \exp(280/TEMP) \times 0.999 : CH_3O_2 + NO = CH_3O + NO_2$   (Eq. R1)

$3.00 \times 10^{-12} \times \exp(280/TEMP) \times 0.001 : CH_3O_2 + NO = CH_3NO_3$        (Eq. R2)

where, TEMP is the temperature, and the numbers of 0.999 and 0.001 are the branching ratio for Eq. R1 and R2, respectively.

When calculating the ozone production and loss in Eq.3, we just marked the RO path in the master chemical mechanism, such as Eq. R1, and excluded the alkyl nitrate branch (Eq. R2). Therefore, the $k_{RO2+NO}$ in Eq. 3 is indeed the reaction rate constant between $RO_2$ and $NO$ that produce the RO and $NO_2$. In the revised manuscript, we clearly clarified it "*$k_{HO2+NO}$ is the second reaction rate between HO₂ and NO; and $k_{RO2+NO}$ is the second reaction rate for the reaction between RO₂ and NO, which only produces RO and NO₂.*" in lines 214-215.

8. Relatedly, in Figure 3, a comparison of panels (c) and (d) appears to indicate that the loss of ozone via reaction of O1D with $H_2O$ increases when using PIC VOCs rather than measured. Please elaborate why this is, is this just because there is more $O_3$ (and so more O1D) in the model with PIC VOCs, and so the rate increases even through the amount of $H_2O$ remains constant? If so, it might help to include a companion figure to Figure 3 in the supplement that shows $P(O_3)$ and $L(O_3)$ as percentages of $O_3$ production/loss to help the reader understand the relative change in loss processes as well.

**Response:** Thank you for your comments and good suggestion. Yes, the loss rate of ozone via reaction of O1D with $H_2O$ ($2.2 \times 10^{-10}$ $cm^3$ molecule$^{-1}$ s$^{-1}$ at 298K, Atkinson et al., 2004) is higher when using the PIC-VOCs than that between ozone and alkenes ($1.27 \times 10^{-17}$ $cm^3$ molecule$^{-1}$ s$^{-1}$ at 298K, Atkinson et al., 1994). Therefore, photolysis of ozone contributes more O1D, leading to enhanced consumption of $O_3$ by $H_2O$. Figure

R9 shows the percentages of different paths of P(O₃) and L(O₃). The relative contributions of the reactions between $O_3$ and alkenes (O1D3) and between $NO_2$ and OH (O1D4) to $O_3$ loss decrease calculated on the basis of the PIC-VOCs when compared with the measured VOCs, while the it increases obviously for other two paths, i.e. O3D1 and O3O2.

[Figure]

**Figure R9.** The percentages of different ozone production and loss rate (observed VOCs in a and c, and PIC-VOCs in b and d). The upper and lower panels present the percentages of different ozone production and loss rate, respectively.

We added the Figure R9 as Figure S13 in the SI. In the revised manuscript in lines 345-349, we added a short paragraph "Figure S13 shows the percentages of the different paths of P(O₃) and L(O₃). The relative contributions of the reactions between $O_3$ and alkenes (O1D3) and between $NO_2$ and OH (O1D4) to the $O_3$ sinks decreased when calculated based on PIC-VOCs compared with those of the measured VOCs, while they obviously increased for the other two paths, i.e., O3D1 and O3O2.".

Minor comments

1. The argument made in lines 69-75 of the introduction about the different mixing ratios of VOCs at the source vs. measurement site is confusing on a first read because it is not clear that the scenario which applies here is where the source one is attempting to control with policy is significantly upwind of the measurement site. If we were considering a source (e.g. traffic) which is closely clustered around the measurement site, then the VOCs measured at the site will be the correct concentrations to consider for $O_3$ production.

**Response:** Thank you for good comments. We agree with you that the VOCs measured at the site will be the correct concentrations to consider for $O_3$ production if we are considering a source (e.g. traffic) which is closely clustered around the measurement site. However, on a regional or urban scale, our argument is reasonable. In the revised manuscript, we have made it clearer "The mixing ratios of VOCs observed at a sampling site are actually the residues of VOCs from emissions due to the photochemical loss during transport from the source site to the receptor site. If photochemically consumed VOCs are not considered, the $O_3$ formation sensitivity and net $O_3$ production may be misunderstood, and subsequent policymaking on $O_3$ pollution prevention at regional or urban scales may be misguided. Thus, the photochemical age-based approach has been applied to evaluate the effect of photochemical processes on VOC measurements (Shao et al., 2011)" in lines 69-75.

2. It is unclear whether Figure 1 and (to a lesser extent) Figure 3 are for one specific day or the entire campaign. For these figures, please specify the time period considered (since Figure 5 is specific to one day).

**Response:** Thank you for your suggestion. Both Figure 1 and Figure 3 are for the entire campaign. In the captions of Figures 1 and 3 in the revised manuscript, we clarified it as "The data range is August 1-28, 2019".

3. For Figure 1, specify which series are measurements and which are calculated. (I assume all but the TVOC PIC series are measurements, but please be explicit.)

**Response:** Thank you. In the revised manuscript, we added a short sentence "The data represent measured results, except for those of the TVOC_PIC, which are calculated based on OH radical exposure." in the caption of Figure 1.

4. For Figure 2, define specific times (i.e. "8 AM" or "11 AM to 1 PM") rather than "morning" and "noon" so that we can compare to Figure 1. Also, line 278 seems to imply that the "noon" points are actually 15:00? That is confusing.

**Response:** Thank you. In the revised manuscript, we have updated the caption of Figure 2 "The pentagons and starts indicate the status in the morning (09:00-10:00) and at noon (14:00-15:00), respectively." In lines 276-279 in the revised manuscript, we also updated the specific time "The black full star and pentagon denote the observed concentrations of NO$x$ and VOCs in the morning (09:00-0:00) and at noon (14:00-15:00), respectively, while the blue symbols are the corresponding values of PICs".

5. Please explain in the caption what the percentages in Figure 4 represent; I only saw a description of the other numbers as production rates in ppb h$^{-1}$. In general, the discussion of the radical chain on lines 362-387 is pretty dense and difficult to follow, but I cannot give any suggestions to improve it without understanding what all the elements in Figure 4 are.

**Response:** Thank you for your good suggestion. The percentages in the box are relative contributions of different reaction paths to the production rate of corresponding radicals in Figure 4. In the revised manuscript, we have updated the sentence in the caption of Figure 4 "The numbers or percentages outside and inside the brackets are the average formation rates (ppb h$^{-1}$) or relative contributions of the corresponding reaction path based on observed VOCs and PIC-VOCs, respectively, to a specific radical.".

In lines 365-390 in the revised manuscript, we have updated this paragraph "The budget of OH-HO$_2$-RO$_2$ radicals was further analyzed to understand the photochemical O$_3$ formation process. The comparison of the radical budget derived from the observed

and PIC-VOCs is shown in Figure 4. The radical cycles are divided into radical sources (green boxes), radical sinks (black boxes), radical propagations (red circles) and equilibria between radical and reservoir species (yellow boxes). The numbers or percentages are the average formation rates (ppb h$^{-1}$) or relative contributions of the corresponding reaction path based on the observed VOCs (outside the brackets) and the PIC-VOCs (inside the brackets) to a certain radical. The relative contributions of different radical paths based on the observed VOCs (outside the brackets) were comparable with those reported in Beijing, Shanghai, and Guangzhou (Tan et al., 2019), while variations were observed for some reaction paths based on the PIC-VOCs. For example, the reaction between ozone and alkenes based on initial VOC concentrations (percentages inside the brackets) contributed more to OH (from 7% to 21%) and HO$_2$ radical production (from 6% to 12%), while photolysis of HONO and HCHO contributed less to the production of OH (from 76% to 60%) and HO$_2$ radicals (from 44% to 40%), respectively. Other radical sources were consistent between the two scenarios. Interestingly, the average formation rates of OH, HO$_2$ and RO$_2$ radicals derived from the PIC-VOCs were obviously higher than those from the observed VOCs. In particular, the oxidation of NO by RO$_2$ and HO$_2$ increased by 1.6 and 1.3 ppb h$^{-1}$, respectively. The enhanced oxidation rate of NO was equal to the increase in the average F(O$_3$) in the analysis process above. This meant that the radical propagation of OH-RO$_2$-HO$_2$ sped up in the case of PIC-VOCs, subsequently accelerating the chemical loop of NO-NO$_2$-O$_3$. For the radical sinks and equilibria related to HNO$_4$, RONO$_2$ and PAN, the values were basically comparable between the two scenarios. In addition, the

O$_3$ formation from the RO$_2$ path increased by 4.1% (from 39.5% to 43.6%) in the simulation using the PIC-VOCs compared with the observed VOCs. The above budget analysis explained the observed increases in F(O$_3$) (~3 ppb h$^{-1}$), which were mainly driven by the reaction of missed reactive VOCs, such as alkenes, with O$_3$.".

Summary

While this paper is a fair study of how one might account for degradation of VOCs between sources and measurements in order to formulate better approaches to controlling O$_3$ production, there have been a number of earlier studies looking at this problem in Beijing. In my opinion, in order for a revision to be considered for publication, the authors must revise the paper to clarify what new information their work adds compared to the previous studies or refocus the paper as a replication study or an update to more recent times. In this second case, the revision should include a thorough comparison with previous studies of this effect in the Beijing area.

**Response:** Thank you for your comment. We have responded all your good comments and suggestions aforementioned. In conclusion, we quantitatively discussed the influence of VOCs degradation between sources and measurements on understanding ozone pollution based on OBM simulations. When comparing with these previous studies estimating OFP with the MIR method, this study has accounted for the non-linear relationship of O$_3$ formation to VOCs and NO$x$. In addition, we provide more details about this issue based on budget analysis of the crucial radicals related to O$_3$ formation. An underestimation should be about 3 ppb h$^{-1}$ or 36 ppb day$^{-1}$ for O$_3$ production rate, which is mainly driven by the reaction between the missed reactive VOCs, such as alkenes, with O$_3$ during our observation. In addition, highly reactive alkenes by the photochemical oxidation accelerated the OH and HO$_2$ radical cycle.

References

Carter, W. P. L.: Development of Ozone Reactivity Scales for Volatile Organic Compounds, J. Air Waste Manage., 44, 881– 899, 1994.

Atkinson R , Baulch D L , Cox R A , et al. Evaluated kinetic and photochemical data for atmospheric chemistry: Volume I - gas phase reactions of Ox, HOx, NOx and SOx species[J]. Atmospheric Chemistry and Physics, 2004, 4(6):1461-1738.

Gao, J., Zhang, J., Li, H., Li, L., Xu, L., Zhang, Y., Wang, Z., Wang, X., Zhang, W., Chen, Y., Cheng, X., Zhang, H., Peng, L., Chai, F., and Wei, Y.: Comparative study of volatile organic compounds in ambient air using observed mixing ratios and initial mixing ratios taking chemical loss into account: A case study in a typical urban area in Beijing, Science of The Total Environment, 628-629, 791:804, 2018.

Li, L., Xie, S., Zeng, L., Wu, R., and Li, J.: Characteristics of volatile organic compounds and their role in ground-level ozone formation in the Beijing-Tianjin-Hebei region, China, Atmospheric Environment, 113, 247:254, 2015.

Shao, M., Lu, S., Liu, Y., Xie, X., Chang, C., Huang, S., and Chen, Z.: Volatile organic compounds measured in summer in Beijing and their role in ground-level ozone formation, Journal of Geophysical Research: Atmospheres, 114, 2008JD010863, 2009.

Shao, M., Wang, B., Lu, S., Yuan, B., and Wang, M.: Effects of Beijing Olympics Control Measures on Reducing Reactive Hydrocarbon Species, Environ. Sci. Technol., 45, 514:519, 2011.

Xie, X., Shao, M., Liu, Y., Lu, S., Chang, C.-C., and Chen, Z.-M.: Estimate of initial isoprene contribution to ozone formation potential in Beijing, China, Atmospheric Environment, 42, 6000:6010, 2008.

Zhan, J., Feng, Z., Liu, P., He, X., He, Z., Chen, T., Wang, Y., He, H., Mu, Y., and Liu, Y.: Ozone and SOA formation potential based on photochemical loss of VOCs during the Beijing summer, Environmental Pollution, 285, 2021.

Roberts, J. M.; Fehsenfeld, F. C.; Liu, S. C.; Bollinger, M. J.; Hahn, C.; Albritton, D. L.; Sievers, R. E. Measurements of aromatic hydrocarbon ratios and NO$x$ concentrations in the rural troposphere: Observation of air mass photochemical aging and NO$x$ removal. *Atmos. Environ.* 1984, *18* (11), 2421–2432.

Bertman, S. B.; Roberts, J. M.; Parrish, D. D.; Buhr, M. P.; Goldan, P. D.; Kuster, W. C.; Fehsenfeld,

F. C. Evolution of alkyl nitrates with airmass age.J.Geophys.Res.-Atmos.1995,*100*(D11), 22805–22814.

Goldan, P. D.; Parrish, D. D.; Kuster, W. C.; Trainer, M.; McKeen, S. A.; Holloway, J.; Jobson, B. T.; Sueper, D. T.; Fehsenfeld, F. C.2000, Airborne measurements of isoprene, CO, and anthropogenic hydrocarbons and their implications. *J. Geophys. Res.-Atmos*, *105* (D7), 9091–9105.

Jobson, B. T.; Berkowitz, C. M.; Kuster, W. C.; Goldan, P. D.; Williams, E. J.; Fesenfeld, F. C.; Apel, E. C.; Karl, T.; Lonneman, W. A.; Riemer, D. Hydrocarbon source signatures in Houston, Texas: Influence of the petrochemical industry. *J. Geophys. Res.-Atmos.* 2004, *109* (D24), D24305.

Lu, K. D., Hofzumahaus, A., Holland, F., Bohn, B., Brauers, T., Fuchs, H., Hu, M., Haeseler, R., Kita, K., Kondo, Y., Li, X., Lou, S. R., Oebel, A., Shao, M., Zeng, L. M., Wahner, A., Zhu, T., Zhang, Y. H., and Rohrer, F.: Missing OH source in a suburban environment near Beijing: observed and modelled OH and HO2 concentrations in summer 2006, Atmospheric Chemistry and Physics, 13, 1057-1080, 2013.

Tan, Z., Fuchs, H., Lu, K., and Hofzumahaus, A.B.,Birger; Broch,Sebastian; Dong,Huabin; Gomm,Sebastian; Haeseler,Rolf; He,Lingyan; Holland,Frank; Li,Xin; Liu,Ying; Lu,Sihua; Rohrer,Franz; Shao,Min; Wang,Baolin; Wang,Ming; Wu,Yusheng; Zeng,Limin; Zhang,Yinsong; Wahner,Andreas; Zhang,Yuanhang;: Radical chemistry at a rural site (Wangdu) in the North China Plain: observation and model calculations of OH,HO2 and RO2 radicals, Atmospheric Chemistry and Physics, 17, 663-690, 2017.

---

## Author Comment (AC2)

Dear Reviewer,

We appreciate your careful consideration of our manuscript. We have carefully responded to all of your **point-by-point** comments and issues and have revised the manuscript accordingly. These revisions are described in detail below.

This manuscript uses a chemical box model approach to calculate VOC and $NO_x$ loss rates, radical chemistry and ozone production rates when constrained with pollutant concentrations in a suburb of Beijing, China. Overall, I recommend revisions of the modelling approach. Below are my specific comments.

**Response:** Thank you for your positive comments and good suggestions. We will respond your comments point-by-point.

1. Line 43. Reference for Seinfeld is missing Pandis author.

**Response:** Thank you for your suggestion. We have updated the missing author in the corresponding position. In addition, we have double checked and updated all references in the revised manuscript.

2. Line 125. To compare the model vs measurements, it is advised to look at slope, intercept and not just correlation coefficient.

**Response:** Thank you for your good suggestion. In the revised SI, we updated the Figure S2 with adding the slope and intercept. We have also updated in the revised manuscript "The correlation coefficient is 0.9 (with a slope of 0.7), indicating that the concentrations of NMHCs are comparable using these two measurement techniques" in lines 124-126.

It should be noted that in line 108, we are discussing the performance between the two instruments for VOC measurements rather than that between the model and measurements. When we comparing the performance of the model with measurements, the slope was also added in addition to the correlation coefficient in the revised manuscript (line 203).

3. Line 147. The hydrocarbon ratio photochemical clock concept works well when you have an isolated source that co-emits the hydrocarbons and a receptor site with no emissions in between. If there are different sources in between and also mixing of airmasses into the plume with different photochemical ages then it complicates the concept and rationale. The use of an early morning time to define the aromatic ratio is also not explained well. The early morning measurements likely reflect the concentration of local emissions under a shallow inversion layer. If the emissions are uniformly distributed across region, then this might be the best time to estimate the emission ratio for xylene to ethylbenzene. The assumption that the emission source remains constant over the region is questionable. It might be helpful to look at gridded regional air quality emissions over the region and plot the emission ratio to see if it is relatively uniform spatially.

**Response:** Thank you for good comments and suggestion. We agree with you that the uniform distribution and constant emissions of VOCs sources are critical for calculating chemical loss of VOCs using hydrocarbon ratio photochemical clock. At least, the emission pattern of VOCs should be constant during the whole day when compared with that in the early morning. Previous studies have found that the ambient ratios of VOCs can reflect their relative emission rates from sources (Golden et al., 2000; Jobson et al., 2004). To verify the rationality of this assumption, Shao et al (2011) had tested relative emission rates from sources by testing different ambient ratios of four pairs of hydrocarbons, i.e., benzene vs acetylene, trans-2-butene vs cis-2-butene, ethene vs toluene and n-hexane vs toluene, that having similar $k_{OH}$ values in each pair. We also further choose the benzene vs acetylene and n-hexane vs toluene to check whether VOCs emissions are constant during our observations. The results are showed in Figure R1 (same as Figure S5). The linear correlation coefficients ($R^2$) were higher than 0.7 and were equal to those reported by Shao et al (2011). This indicates that the assumption of a constant VOCs emission should be reasonable during our observations.

[Figure]

**Figure R1.** The relationship between the concentration of toluene vs acetylene and n-hexane vs toluene.

As for the assumption that the emission source remains constant over the region, it is better to look at gridded regional air quality emissions over the region. Unfortunately, such regional gridded data are unavailable at the present time. Alternatively, we performed spatial distribution analysis using a source-receptor model (potential source contribution function, PSCF). In Figure R2 (same as Figure S6), the emissions of VOCs are not spatially uniform in Beijing, i.e., strong emissions are in the south or southeast directions. However, the PSCF patterns in the daytime are highly similar to that in the early morning. This means that the emissions of VOCs should be constant during the daytime.

[Figure]

**Figure R2.** The potential source contribution function (PSCF) maps for the ratio of xylene to ethylbenzene (a and b), ethylbenzene (c and d), and xylene (e and f) arriving in the observation site. The figures of a, c and e are the results of 05:00 and 06:00, and the figures of b, d and f are the results of daytime (07:00-19:00).

In the revised manuscript, we added the sentences ". In previous work (Shao et al., 2011; Zhan et al., 2021), the selection of ethylbenzene and xylene as tracers was justified for calculating ambient OH exposure under the following conditions: 1) the concentrations of xylene and ethylbenzene were well correlated (Figure S4), which indicated that they were simultaneously emitted; 2) they had different degradation rates in the atmosphere; and 3) the calculated PICs were in good agreement with those calculated using other tracers, such as i-butene/propene (Zhan et al., 2021). To test the

relative constant emission ratio from different sources, we chose benzene vs. acetylene and n-hexane vs. toluene as references, and the result is shown in Figure S5. These ambient ratios could directly reflect their relative emission rates from sources (Goldan et al., 2000; Jobson et al., 2004). The linear correlation coefficients ($R^2$) were generally higher than 0.7, which were equal to that reported by Shao et al. (2011). To further test the assumption that the emissions of xylene and ethylbenzene were constant throughout the day, their potential sources were calculated using a source-receptor model (the potential source contribution function, PSCF). As shown in Figure S6, xylene and ethylbenzene showed similar distributions. In addition, the ratio of ethylbenzene/xylene at 5:00 and 6:00 was similar to that during the daytime. These results indicated that the emissions of xylene and ethylbenzene were constant throughout the day." in lines 160-175. In future, we will try to combine OBM and the regional air quality model to better understand the influence of photochemical initial VOCs.

4. Line 178-182. The mean daytime OH used for the calculation was 4.3E6 molec/cm$^3$. Please state that the measured aromatic ratio was also a mean value for the same daytime period used to calculate the OH.

**Response:** Thank you for good comments. Indeed, here we are discussing the daytime OH exposure (molecules cm$^{-3}$ h) but not OH concentration (molecules cm$^{-3}$). We calculated hourly OH exposure from 07:00 to 19:00 using the ratio of xylene to ethylbenzene according to Eqs.1 and 2 in the manuscript. Then, we obtained the mean daytime OH exposure 4.3±1.9 molecules cm$^{-3}$ h. The mean daytime OH concentration (4.3±1.9×10$^6$ molecules cm$^{-3}$) was calculated based on the JO1D using the method reported in our previous work (Liu et al., 2020b, Liu et al., 2020c). In lines 178-182 in the revised manuscript, we rephrased the sentences to make it clearer "The OH exposure was from 0.82 to 8.1×10$^6$ molecule cm$^{-3}$ h, with a mean daytime value of 4.3±1.9×10$^6$ molecules cm$^{-3}$ h. Accordingly, the mean photochemical ages were 1.7±0.9 h using the mean daytime (8:00-17:00 LT) OH concentrations (4.3±3.1×10$^6$ molecules cm$^{-3}$) calculated based on JO1D using the method reported in our previous work (Liu

et al., 2020b; Liu et al., 2020c).".

5. Line 185. I agree that the biogenic isoprene emission is not co-located with the aromatic emission sources and this complicates the ratio method. Can the authors look at regional model emissions for isoprene to see if the isoprene sources are local near the site or whether they are closer to the aromatic sources? A possible assumption is that the production and loss of isoprene balance along the transport and concentration of isoprene remain constant in the trajectory from the aromatic source region to the site. Again, regional air quality model results would be able to show what the distribution of isoprene looks like around the site.

**Response:** Thank you for good suggestion. We think that isoprene is mainly from biogenic emissions in summer but not co-located with the aromatic emission sources. This is evidenced by the diurnal variations of isoprene, xylene and ethylbenzene (Figure R3, same as Figure S14) although the regional model emissions for isoprene are unavailable at the present time. As shown in Figure R3, isoprene and aromatics (xylene and ethylbenzene) showed totally different diurnal variations. This means isoprene sources are from biogenic emissions.

[Figure]

**Figure R3.** The mean diurnal curves of xylene, ethylbenzene and isoprene.

On the other hand, like xylene and ethylbenzene, we also carried out PSCF analysis (Figure R4, same as Figure S15) for isoprene to check whether the production and loss of isoprene is balanced along the transport and concentration of isoprene remain constant in the trajectory from the aromatic source region to the site. The results showed that the spatial pattern of isoprene is even during our observations. This indirectly indicated that it could be considered as the balance on production and loss of isoprene along the transport, and the concentration of isoprene remain constant in the trajectory from the aromatic source region to the site.

[Figure]

**Figure R4.** The potential source contribution function (PSCF) maps for the isoprene arriving in the observation site. The figures of a and b are the results of 05:00 and 06:00, and the daytime (07:00-19:00), respectively.

In lines 124-130 in the revised SI, we added the sentences "Isoprene is mainly from biogenic emissions but not co-located with the aromatic emission sources in summer, which is evidenced by the diurnal variations of isoprene, xylene and ethylbenzene (Figure S14) although the regional model emissions for isoprene are unavailable at the present time. The results of PSCF analysis showed that the spatial pattern of isoprene is even during our observations in S15, which indirectly indicated that it could be considered as the balance on production and loss of isoprene along the transport, and the concentration of isoprene remain constant in the trajectory from the aromatic source region to the site.".

6. Line 233. I would recommend "grouped into lumped species" instead of parcels.

**Response:** Thank you for good suggestion. We have updated "the observed VOCs are grouped into different lumped species according to the classification of RACM2" in the lines 233-235 in the revised manuscript.

7. Line 238. Five minutes is enough time for the radicals to reach steady-state but not the NO$x$ and OVOCs. For example, PAN has a long lifetime and would not reach steady state in 5 min. Given that the intent is to "correct" the VOCs to an initial condition and you expect an ~ 2hr photochemical time then an equivalent time to run the model might be best approach and would give time for OVOCs to spin up to more reasonable mixing ratios.

**Response:** Thank you for comments and suggestion. The time interval for the outputs is usually set to 60 min (Lu et al., 2013; Lu et al., 2017; Tan et al., 2019) determined by the time resolution of observation data. However, the time interval of 60 min is too coarse to run the model because the sun can move a lot within one hour. Therefore, a higher time resolution (5 min) is usually taken along with interpolation of

the inputs in OBM simulations to reduce the influence of great distortion of meteorological parameters in a long-time interval (Tan et al., 2017). The time interval (60 min) in this study is similar with that used by Lu et al (2013) and Lu et al (2017), and higher than that (30 min) reported by Tan et al (2018), which indicated that the photochemical time setting was acceptable. We agree with you that OVOCs and PAN have a long lifetime compared with the radicals (OH, $HO_2$ and $RO_2$). Although the long lifetime species would not reach steady state in 5 min, the model outputted their concentrations with a time resolution of 60 min. This indeed accounted for the time for OVOCs and PAN to spin up to more reasonable mixing ratios. In addition, the chemical cycle of radicals is critical for ozone formation. The simulation is more reliable once the radical sources, sinks and propagation of radicals can be well simulated at this time interval.

In the revised manuscript, we have updated the sentences "The chemical model simulated photochemical reactions with input species for a time interval of 60 minutes, which was enough for $NOx$, OH, $HO_2$, and $RO_2$ to reach a steady state because the typical relaxation time of the chemical system is 5-10 minutes in summer (Tan et al., 2018). However, all the species and parameters were input at a 5 min interval by data interpolation to reduce simulation inconsistencies and large distortions of meteorological parameters at longer time intervals (Tan et al., 2018)" in lines 235-240.

8. One major concern that I have in the photochemical initiation of VOCs is that the authors are not considering the correction needed for $NOx$ as it also reacts in the trajectory reaching the site. The VOCs and $NOx$ can be emitted by common combustion processes, particularly in an urban area. The authors also note they follow the same diurnal profile as VOCs. The lifetime of $NOx$ is comparable to some VOCs so why not correct the $NOx$ as well? Other studies have done this and in fact the $NOx$/$NOy$ is an alternative ratio in the photochemical clock method (Hayes et al., 2013).

**Response:** Thank you for good comments and suggestion. We agree with you that photochemical conversion of $NOx$ should be corrected. Because the $NO_y$ concentrations were unavailable during our observations, we simply corrected the

concentration of NO and $NO_2$ using the OH exposure and assuming a gas phase oxidation by OH radical ($k_{NO-OH}$=3.0×10$^{-11}$ cm$^{-3}$ molecules s$^{-1}$ and $k_{NO2-OH}$=1.2×10$^{-11}$ cm$^{-3}$ molecules s$^{-1}$, Atkinson et al., 2004). The corrected data are showed in Figure R5. The average concentrations of NO, $NO_2$, and $NOx$ were underestimated 0.6±0.6 ppb, 1.5±1.4 ppb, and 2.1±2.0 ppb, respectively, lower than the underestimated VOCs (6.9±0.5, Figure 1), which indicated that the role of the photochemical loss of $NOx$ is less important than VOCs when evaluating the ozone formation process and mechanism. In the early morning, the PICs of $NOx$ were almost the same as the observed values. On the other hand, the previous studies (Li et al., 2020; Tan et al., 2019; Sun et al., 2011) have showed that the Beijing belonged to VOCs-limited regime, which means that the $NOx$ is sufficient during the nonlinear relationship of $NOx$-VOCs-$O_3$. In other word, the chemical loss of $NOx$ compared with photochemical loss of VOCs was less important in Beijing. Therefore, we focus on the contribution of photochemical loss of VOCs to ozone formation mechanism in this study. And in future, we will compare the ratio of $NOx$ to $NO_y$ (alternative calculation) with the ratio method of xylene to ethylbenzene.

[Figure]

**Figure R5.** The mean diurnal curves of NO, $NO_2$ and $NOx$. The corrected concentration of NOx was calculated with the OH exposure based on the ratio method.

9. Line 248. If I understand this section, the observed ozone max was 119 ppbv and the modeled max was 70 ppbv. This seems like a large difference. If this can be explained by the mixing and transport of ozone from regional scale then maybe it might be best to constrain the ozone to the observed value each hour. For diagnosing the chemistry pathways (Figure 3,4), this ensures the ozone is at a reasonable level for calculating the alkene ozonolysis and O1D+H2O reaction rate.

**Response:** Thanks for your good comments and suggestion. The mean ozone concentration during the observation period was 44.8±27.2 ppb with a maximum of 119.1 ppb (Figure S5). As shown in Figure S5, the modeled $O_3$ concentrations actually were comparable with the observed values. It is worth to note that we used the average data of the campaign as input to simulate the EKMA curve in this study, and then we set up 30×30 matrixes by reducing or increasing the measured VOCs and $NOx$ concentrations to the model input according to the base case. The modeled max value you mentioned (70 ppb in Figure 2) is comparable with the maximal mean ozone concentrations during our observations (Figure 1). When we diagnosing the chemistry pathways (Figure 3 and 4), we have constrained $O_3$ concentrations. Thus, this ensures the ozone is at a reasonable level for calculating the reaction rates between alkene and ozonolysis, and between O1D and $H_2O$.

In the revised manuscript, we have added a sentence "It is worth mentioning that the average survey data were selected as the baseline scenario in simulating the EKMA curve in this study." in lines 241-242.

10. Line 313. The daytime average $P(O_3)$ is calculated at 3 ppbv/hr higher than with using measured VOCs. The authors appear to extrapolate to a 24-hr average by multiplying by 24 hr to get 36 ppbv/hr higher than with the measured VOCs. The nighttime $P(O_3)$ averages could be different than daytime averages, so it would be preferred to state what the daytime average difference is between corrected and

measured VOC approaches.

**Response:** Thanks for your good comments. The OBM in this study was just used to simulated the daytime photochemical process (07:00~19:00 LT). Thus, the daytime average of $F(O_3)$ is 3.0±2.1 ppb h$^{-1}$ higher than with using measured VOCs. And the value of ~36 ppb day$^{-1}$ is just calculated from 07:00 to 19:00 based on the average $F(O_3)$, not including the nighttime.

In the revised manuscript, we updated the sentences "The average daytime $P(O_3)$ from 07:00 to 19:00 based on the initial concentrations of VOCs was 4.0±3.1 ppb h$^{-1}$ higher than that based on the measured VOCs concentrations (Figure 3b). At the same time, the $F(O_3)$ from 07:00 to 19:00 based on the initial concentrations of VOCs was also 3.0±2.1 ppb h$^{-1}$ higher than the measured counterpart (Figure S11). Thus, the net $O_3$ production could be accumulatively underestimated by ~36 ppb day$^{-1}$ from 07:00 to 19:00 if the consumption of VOCs was not considered." in lines 310-315.

11. Line 432. I think the discussion of the diurnal profile and the different chemical and transport processes should include the mixing of the stable nocturnal surface layer in the mid-morning. As the sun heats the surface, there is significant mixing of surface layer with air above in the residual atmospheric layer which is likely composed air from the prior day mixed boundary layer and then transported to the site overnight. This residual air likely contains hydrocarbons and ozone from different sources and at different photochemical ages. The ozone increase around 9-10am is often associated with this vertical mixing. The ozone mixing down to surface is photochemically produced but from a different region from previous days (unless the region is influenced by a lake/land breeze where recirculation of the same air mass can occur.

**Response:** Thanks for your good comments. We agree with you that the diurnal profile of ozone will be affected by the mixing of the stable nocturnal surface layer in the mid-morning. It should mention that $R(O_3)$ is the physical transportation including horizontal and vertical mixing, and we attributed the variation of ozone in the morning all to the transportation. In fact, the upper layer is known as the residual layer (RL) in the morning, which is isolated from the surface due to inversion at night (Tan et al.,

2021). RL usually contains the air mass with higher $O_3$ concentration than that of nocturnal boundary layer, and when the boundary layer is gradually uplifted, the vertical transport was important due to the fast entrainment.

In the revised manuscript, we have added the sentences "This was mainly because the residual layer (RL) that formed at night was unfavourable for the inversion of airmass in the early morning (Tan et al., 2021). The RL usually contains an airmass with a higher concentration of $O_3$ than that in the nocturnal boundary layer. Vertical transport becomes prominent due to the fast entrainment when the boundary layer is gradually uplifted." in lines 433-437.

12. Line 388. The authors state that "The radical budget analysis illustrated that the $O_3$ formation processes between the observed and photochemical initial VOCs showed no significant difference." The title of the manuscript implies that the initial photochemical loss of VOCs does have an impact on the ozone formation mechanism. It seems that the ozone production rates are sensitive to the corrected VOC loss (and likely corrected NO$x$ loss as well). Maybe an improved title could be "Influence of Photochemical Loss of VOCs and NOx on Ozone Formation Rates and Diagnosed Ozone Production Sensitivity in Beijing, China"

**Response:** Thank you for comments and suggestion. As above, the chemical loss of NO$x$ compared with photochemical loss of VOCs was less important in Beijing, and we focus on the contribution of photochemical loss of VOCs to ozone formation mechanism in this study. Actually, the current title "Influence of Photochemical Loss of VOCs on Understanding Ozone Formation Mechanism" might also reflect what you mean. However, we still prefer the original one to make it more concise.

13. Figure S9 shows some days with large model over-predictions. Can the authors explain what factors are contributing to the model over-predictions?

**Response:** Thank you for good comments. The OBM is zero-dimensional atmospheric modeling coupled with just gas-phase reaction, which means that it cannot

consider the influence of meteorological process (vertical diffusion and horizontal transportation). The dilution caused by wind or boundary layer was not well represented during the simulation, and this may lead overestimation or underestimation to some extent. The similar over-predictions were also reported by Zong et al (2018) and Zhang et al (2020), but it cannot affect the simulation of ozone formation process and mechanism because we have the constrained ozone concentration during simulations. In the revised manuscript, we added the sentences "It is worth mentioning that the results of model simulation can sometimes be overestimated or underestimated to some extent, which has also been reported by previous studies (Zong et al., 2018; Zhang et al., 2020), but this did not affect our simulations of the ozone formation process and mechanisms because we constrained the ozone concentration during our simulations." in lines 204-208.

14. Overall, I really like Figures 3, 4 and 5. My recommendation would be to include a NO$x$ correction, as well as the VOC correction, and to consider a longer model time so that the OVOCs reach closer to their typical ambient concentrations. Of particular interest would be the aromatic and monoterpene oxidation products as the precursor aromatic and monoterpene have an intermediate lifetime (several hours) and their OVOCs are not typically measured. The figures are of very good quality. The paper needs some improvements for English language.

**Response:** Thank you for your positive comments and good suggestions. We have carefully responded to all of your point-by-point comments and issues above and have revised the manuscript and SI accordingly. We also carefully corrected language errors by a native speaker (as shown in Figure R6). The corrections have been marked in blue in the revised manuscript.

[Figure]

**Figure R6.** The editing certificate by the highly native English speaking editors at

AJE.

References:

Shao, M., Wang, B., Lu, S., Yuan, B., and Wang, M., 2011. Effects of Beijing Olympics Control Measures on Reducing Reactive Hydrocarbon Species, Environmental Science & Technology, 45, 514-519.

Goldan, P. D.; Parrish, D. D.; Kuster, W. C.; Trainer, M.; McKeen, S. A.; Holloway, J.; Jobson, B. T.; Sueper, D. T.; Fehsenfeld, F. C.2000, Airborne measurements of isoprene, CO, and anthropogenic hydrocarbons and their implications. *J. Geophys. Res.-Atmos*, *105* (D7), 9091–9105.

Jobson, B. T.; Berkowitz, C. M.; Kuster, W. C.; Goldan, P. D.; Williams, E. J.; Fesenfeld, F. C.; Apel, E. C.; Karl, T.; Lonneman, W. A.; Riemer, D. Hydrocarbon source signatures in Houston, Texas: Influence of the petrochemical industry. *J. Geophys. Res.-Atmos.* 2004, *109* (D24), D24305.

Zhang, Q., Li, L., Zhao, W., Wang, X., Jiang, L., Liu, B., Li, X., and Lu, H.: Emission characteristics of VOCs from forests and its impact on regional air quality in Beijing, China Environmental Science, 41, 622-632, 2021b.

Lu, K. D., Hofzumahaus, A., Holland, F., Bohn, B., Brauers, T., Fuchs, H., Hu, M., Haeseler, R., Kita, K., Kondo, Y., Li, X., Lou, S. R., Oebel, A., Shao, M., Zeng, L. M., Wahner, A., Zhu, T., Zhang, Y. H., and Rohrer, F.: Missing OH source in a suburban environment near Beijing: observed and modelled OH and HO2 concentrations in summer 2006, Atmospheric Chemistry and Physics, 13, 1057-1080, 2013.

Lu, X., Chen, N., Wang, Y., Cao, W., Zhu, B., Yao, T., Fung, J. C. H., and Lau, A. K. H.: Radical budget and ozone chemistry during autumn in the atmosphere of an urban site in central China, Journal of Geophysical Research-Atmospheres, 122, 3672-3685, 2017.

Tan, Z., Lu, K., Jiang, M., Su, R., Wang, H., Lou, S., Fu, Q., Zhai, C., Tan, Q., Yue, D., Chen, D., Wang, Z., Xie, S., Zeng, L., Zhang, Y., 2019. Daytime atmospheric oxidation capacity in four Chinese megacities during the photochemically polluted season: a case study based on box model simulation. Atmos. Chem. Phys. 19, 3493-3513.

Tan, Z., Lu, K., Jiang, M., Su, R., Dong, H., Zeng, L., Xie, S., Tan, Q., and Zhang, Y.: Exploring ozone pollution in Chengdu, southwestern China: A case study from radical chemistry to O-3-VOC-NOx sensitivity, Science of the Total Environment, 636, 775-786, https://doi.org/10.1016/j.scitotenv.2018.04.286, 2018.

Li Q , Su G , Li C , et al. An investigation into the role of VOCs in SOA and ozone production in

Beijing, China[J]. The Science of the Total Environment, 2020, 720(Jun.10):137536.1-137536.14.

Sun, Y., Wang, L., Wang, Y., Quan, L., Zirui, L., 2011. In situ measurements of SO2, NOx, NOy, and O3 in Beijing, China during August 2008. Sci. Total Environ. 409, 933-40.

Zhang K , Li L , Huang L , et al. The impact of volatile organic compounds on ozone formation in the suburban area of Shanghai[J]. Atmospheric Environment, 2020, 232:117511.

Zong R H, Xue L K, Wang T , Wang W X. Inter-comparison of the Regional Atmospheric Chemistry Mechanism (RACM2) and Master Chemical Mechanism (MCM) on the simulation of acetaldehyde[J]. Atmospheric Environment, 2018, 186:144-149.

Hayes, P. L., Ortega, A. M., Cubison, M. J., Froyd, K. D., Zhao, Y., Cliff, S. S., Hu, W. W., Toohey, D. W., Flynn, J. H., Lefer, B. L., Grossberg, N., Alvarez, S., Rappenglück, B., Taylor, J. W., Allan, J. D., Holloway, J. S., Gilman, J. B., Kuster, W. C., de Gouw, J. A., Massoli, P., Zhang, X., Liu, J., Weber, R. J., Corri[1]gan, A. L., Russell, L. M., Isaacman, G., Worton, D. R., Kreis[1]berg, N. M., Goldstein, A. H., Thalman, R., Waxman, E. M., Volkamer, R., Lin, Y. H., Surratt, J. D., Kleindienst, T. E., Of[1]fenberg, J. H., Dusanter, S., Griffith, S., Stevens, P. S., Brioude, J., Angevine, W. M., and Jimenez, J. L.: Organic aerosol composition and sources in Pasadena, California during the 2010 CalNex campaign, J. Geophys. Res.-Atmos., 118, 9233–9257.

---

## Author Response (AR2)

Dear editor,

We appreciate the careful consideration of our manuscript by the reviewers. We have carefully responded to all of the point-by-point comments and issues raised by the reviewers and have revised the manuscript accordingly. These revisions are described in detail below.

Review 1 #

Thank you to the authors for carefully considering my questions in the first round and thoroughly revising their manuscript. I have a few minor suggestions to clarify the added content for readers, otherwise this paper is ready for publication.

**Response:** Thank you for your positive comments and kind help.

1. In the expanded literature review in the introduction (around l. 87) or the next paragraph, it may help the reader if the authors enumerated how the MIR method differs from the method used here.

**Response:** Thank you for your good suggestions. The MIR of VOCs is determined according to,

$$MIR = \lim_{\Delta VOC \to 0} \left[ \frac{O_3(VOCs_{MIR}+\Delta VOCs)-O_3(VOCs_{MIR})}{\Delta VOCs} \right] \text{ (Eq. R1)}$$

where, $O_3(VOCs_{MIR})$ and $O_3(VOCs_{MIR}+\Delta VOCs)$ are the simulated maximal $O_3$

concentrations with the VOCs concentration same as that under the MIR conditions and with an increased quantity of VOCs ($\Delta VOCs$), respectively. Therefore, the first step is to determine the MIR conditions including NO concentration and other input parameters of the base scenarios in a specific region. Figure R1 shows the schematic of the VOCs' incremental reactivity (IR) as a function of NO concentration. Thus, the NO

concentration under MIR conditions is the value that corresponds to the maximal IR.

Other input parameters include the concentrations of VOCs, $O_3$, $SO_2$, CO, and HONO, and the meteorological parameters in the base scenarios are usually selected according to their median or mean values in ozone pollution events. Therefore, the MIR values of

VOCs depend on the meteorological conditions, the components of VOCs, and the concentrations of other pollutants even if the O₃ formation is VOCs-sensitive as required by the MIR method. In this work, we performed OBM simulations when discussing the influence of photochemical loss of VOCs on ozone formation. Thus, it reflected the real atmospheric conditions during our observations.

[Figure]

Figure R1. Schematic of VOCs' incremental reactivity as a function of NO

concentration.

We added the sentence "In addition, the MIR values of VOC species for a specific region are calculated with the base scenario, in which NO concentration and other parameters are the values that correspond to the maximal incremental reactivity (IR).

The fixed MIR values of different VOCs can neither reflect the non-linear relationship between ozone and VOCs, involving in the complicated radical recycling (OH-RO₂-

RO-HO₂-OH) related to the production of ozone, nor be used for analyzing the radical budget of the initial VOCs concentration. Thus, a quantitative analysis is necessary to explicitly understand the influence of photochemical loss of VOCs on ozone formation and its mechanisms based on OBM studies, in which the dynamic atmospheric and meteorological conditions is accounted for." in lines 93-102 in the revised manuscript.

2. I didn't see Fig. R3 from the response in the SI. I thought this figure was useful both as a check on the usability of the xylene/ethylbenzene clock and the uncertainty due to the choice of clock, and recommend the authors include it in the supplement.

**Response:** Thank you for your good suggestions. We have added this figure (Figure

R2) as Figure S5 in the revised SI and updated the sentences "3) the calculated PICs were in good agreement with those calculated using other tracers, such as i- butene/propene (Figure S5) (Zhan et al., 2021)." in lines 172-174 in the revised manuscript.

[Figure]

**Figure R2.** Comparison of PICs calculated for xylene/ethylbenzene and i-

  Butene/Propene (Zhan et al., 2021). Error bars are standard deviations.

3. For the comparison between the NMS and OS sites, was there a reason why the comparison could only be done for 1 day? If you could use a larger subset of the campaign, you could get a better statistical distribution of the error on this method. If there are clear reasons why this is the only viable day (e.g. the winds never placed the

NMS site downwind of the OS site on other days), then it would be good to state that.

**Response:** Thank you for your comments. We agree with you that a larger subset will better present the statistical distribution of the error on this method. However, after we checked the wind trajectory of the whole campaign, no other days could match the two sites well. This requires longer observations in the future.

4. Following #3, I recommend adding this comparison to the conclusion, as that is an important piece of information for readers to take away.

**Response:** Thank you for your good suggestions. According to your suggestion, we added the sentence in lines 466-469 "And the mean ozone concentration of downwind site was 27.6 ppb day$^{-1}$ higher than the observation site, slightly lower than the difference (~36 ppb day-1) between PIC-VOCs and observed VOCs, which indirectly supported the accuracy of the above results." in the revised manuscript.

5. In lines 230-231, what is meant by "second reaction rate"? Does this mean the second of two possibilities (e.g. $NO + RO2 \rightarrow NO2 + RO$ vs. $NO + RO2 \rightarrow RONO2$) or second order reactions (i.e. reactions with two reactants)? If the former, I recommend a different way to describe the reactions, as the order in which reactions are listed isn't consistent in every source.

**Response:** Thank you for your good suggestions. The "second reaction rate" is the second-order reaction rate constant. We have corrected this to "the second-order reaction rate constants" in lines 230-231 in the revised manuscript. We also checked the entire text.

Review 2 #

The manuscript has been much improved with the revisions. I feel the reviewer comments were thoughtfully considered and addressed where possible. The areas of uncertainty remaining have been discussed adequately.

**Response:** Thank you for your positive comments and kind help.

I would recommend rephrasing the revised lines 441-446 with the help of a native English speaker. Maybe, these sentences could be considered. "This was mainly because, under stable conditions, the nighttime residual layer (RL) is isolated from mixing with the nighttime surface layer. The RL layer usually contains an air mass with a higher ozone mixing ratio than in the surface layer. In the morning, surface heating causes mixing upward in the surface layer until the temperature inversion is eroded away and rapid mixing of pollutants throughout the surface and boundary layer occurs."

**Response:** Thank you for your good comments and suggestions. We have rephrased the sentences according to your suggestion in lines 441-446 in the revised manuscript "This was mainly because, under stable conditions, the nighttime residual layer (RL) is isolated from mixing with the nighttime surface layer. (Tan et al., 2021). The RL layer usually contains an air mass with a higher ozone mixing ratio than in the surface layer. In the morning, surface heating causes mixing upward in the surface layer until the temperature inversion is eroded away and rapid mixing of pollutants throughout the surface and boundary layer occurs.".

---

## Author Response (AR3)

Dear editor,

  We appreciate the careful consideration of the publication of our manuscript. As for your notice about the information in Figure S1, we have added the source of the map, and the picture in Figure S1 is our own.

  Sincerely

  Yongchun Liu